# ULTRA-LOW ACCUMULATION PRECISION INFERENCE WITH BLOCK FLOATING POINT ARITHMETIC

## ABSTRACT

Block Floating Point (BFP) quantization offers a hardware-efficient numerical range trade-off. Previous studies have quantized weights and activations to an extremely low precision using the BFP arithmetic. However, as the precision of weights and activations diminishes, we identify that accumulation becomes a hardware bottleneck in the BFP MAC. Nevertheless, existing attempts to decrease the precision of accumulation in matrix multiplication generally preserve model performance through training with a pre-selected, fixed accumulation precision. Nonetheless, selecting an unduly low precision leads to notable performance degradation, and these studies lack an effective approach to establish the lower precision limit, potentially incurring considerable training costs. Hence, we propose a statistical method to analyze the impact of reduced accumulation precision on the inference of deep learning applications. Due to the presence of fixed-point accumulation and floating-point accumulation in BFP matrix multiplication, we formulate a set of equations to relate the data range of fixed-point multiply-accumulate operations and the effects of floating-point swamping to the parameters of BFP quantization, the length of accumulation, model weights, and the minimum number of bits required for accumulation, thereby determining the appropriate accumulation precision. Applied to MMLU Llama2-7B, SQuAD-v1.1 BERT-Large and BERT-Base and CIFAR-10 ResNet-50, our precision settings yield performance close to the FP32 baseline. Meanwhile, further precision reduction degrades performance, indicating our approach's proximity to precision limits. Guided by our equations, the hardware exhibits a 13.7%-28.7% enhancement in area and power efficiency over high-precision accumulation under identical quantization configuration, and it demonstrated a $10.3\times$ area reduction and an $11.0\times$ power reduction compared to traditional BFP implementations.

## 1 INTRODUCTION

Deep learning technology has achieved significant success in a wide range of applications through the training of large-scale deep models with extensive datasets. Concurrently, this approach has imposed substantial storage and computational burdens. Quantization emerges as a promising method to reduce the cost of deep learning by diminishing the bit-width of data flow within models, thereby reducing storage and computational overhead (Deng et al., 2020). As an effective numerical system for deep learning, Block Floating Point (BFP) strikes a favorable balance between dynamic range and hardware cost (Drumond et al., 2018). Specifically, previous studies have demonstrated that low-precision BFP formats can achieve accuracy comparable to FP32 under various deep learning workloads (Darvish Rouhani et al., 2020; Drumond et al., 2018; Soloveychik et al., 2022; Köster et al., 2017; Zhang et al., 2022). However, it is observed that as the quantization precision decreases, accumulation becomes a hardware bottleneck in BFP MAC. As illustrated in Figure 1(b), the area occupied by the accumulation component accounts for 17.8%, 33.7%, and 64.4% for BFP16, BFP8, and BFP4, respectively. Therefore, reducing accumulation precision can further enhance hardware efficiency on top of lowering quantization precision.

In BFP MAC, both fixed-point and floating-point accumulations are present. For fixed-point accumulation, a decrease in precision is accompanied by an increased likelihood of overflow. Previous works have focused on avoiding overflow occurrences or mitigating their impact (Colbert et al., 2023; Ni et al., 2020; Xie et al., 2020; Li et al., 2022). Nevertheless, methods to mitigate the impact

of overflow are not guaranteed to maintain accuracy when overflows occur frequently. Hence, we employ the $3\sigma$ principle to predict data ranges and select accumulation precision to prevent overflow permanently. For floating-point accumulation, the phenomenon of swamping (Higham, 1993) becomes more pronounced as precision decreases. Previous work has attempted to correlate the numerical precision loss and model performance degradation due to swamping through variance (Wang et al., 2018; Sakr et al., 2019). Alternatively, our research centers on the inference phase, where we leverage the Frobenius norm(Suh et al., 2022; Yuan et al., 2020) to gauge matrix similarity before and after precision reduction in accumulation. Grounded in the Frobenius norm, we propose the metric Frobenius norm retention rate ($FnRR$) to quantify the degree of swamping resulting from reduced floating-point mantissa precision. Furthermore, we derive a formula $f(n)$ from $FnRR$ to assess the impact of data precision loss on model performance, establishing a connection between floating-point accumulation accuracy and model performance.

Utilizing the derived formula for $FnRR$, our analysis identifies accumulation length as the pivotal factor influencing floating-point accumulation precision. Leveraging this insight, we introduce a segmented accumulation approach to mitigate precision loss. Experimental validation affirms the method's efficacy across diverse model and quantization paradigms. Furthermore, integrating the theoretically deduced precision into hardware yields a 13.7–28.7% reduction in area and power relative to high-precision accumulation under identical quantization conditions, and nearly a $10\times$ enhancement in area and power efficiency compared to FP32 accumulation in BF16 MAC operations.

Our research contributes both theoretical and practical insights. Firstly, we present a theoretical framework for determining the minimum fixed-point accumulation bit-width, emphasizing overflow avoidance based on variance and mean. Secondly, we introduce the $FnRR$ and $f(n)$ metrics to link floating-point accumulation precision with model performance. Our analysis shows that accumulation length is a key determinant in precision selection. To further reduce precision, we employ a segmented accumulation technique. We then validate the accumulation precision boundary through experiments. Finally, we design BFP multiply-accumulators within the established boundaries and assess the improvements in area and power efficiency.

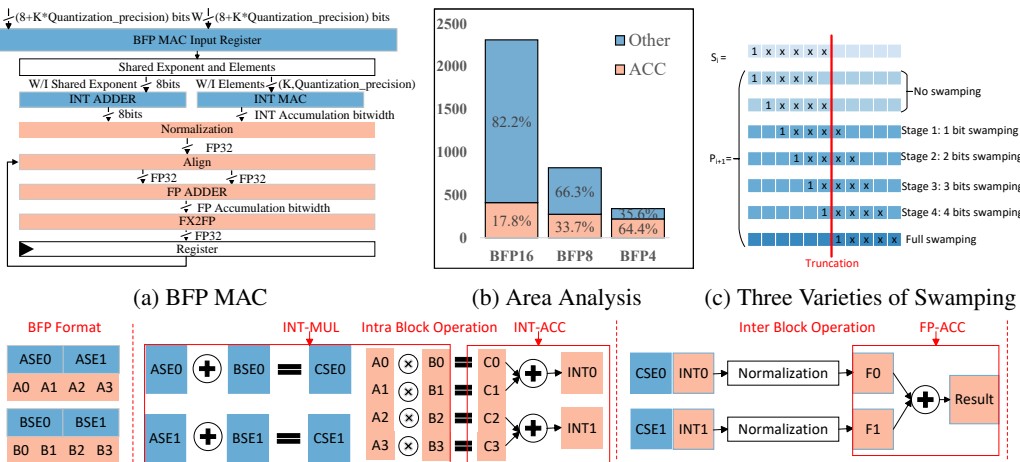

(a) BFP MAC  (b) Area Analysis  (c) Three Varieties of Swamping

(d) BFP format and BFP matrix multipication

Figure 1: (a)A schematic diagram of the BFP MAC unit, (b)Area Analysis of Baseline BFP-MAC. (c)The distinctions among three swamping phenomena When $m_{acc} = 5$ and $m_p = 4$. (d) illustrates a simple demonstration of the data flow in BFP matrix multiplication with a block size of 2.

## 2 RELATED WORK AND BACKGROUND

### 2.1 RELATED WORK

Our work endeavors to establish a theoretical framework for determining the boundary of accumulator bit-width for the BFP format. Although this topic has not been previously discussed, there has been extensive exploration of fixed-point accumulator bit-width and floating-point accumulator bit-width.

**Fixed-point accumulator bit-width** WrapNet (Ni et al., 2020) leverages the cyclic nature of integer computer arithmetic by inserting a differentiable cyclic activation function, rendering neural networks robust to integer overflow. This allows for the selection of ultra-low-precision fixed-point accumulator bit-width. However, they also note that high overflow rates can lead to training instability. A2Q (Colbert et al., 2023) adheres to the principle of avoiding overflow and approach the determination of fixed-point accumulator bit-width boundaries from both the data type and weight perspectives. Xie et al. introduce a quantization range mapping factor $\alpha$ to maximize data representation capabilities while avoiding overflow under a specified accumulator bit-width (Xie et al., 2020). While their training method can ensure model accuracy at an appropriate accumulator bit-width, they do not provide an efficient approach to determine the boundary of the accumulator bit-width.

**Floating-point accumulation bit-width** Wang et al. illustrate that the phenomenon of swamping significantly limits the potential for reducing accumulation precision (Wang et al., 2018). To address this issue, they propose two novel techniques: chunk-based accumulation and floating-point stochastic rounding. These methods allow for the training of Deep Neural Networks (DNNs) even when the accumulation bit-width is decreased to FP16, thereby circumventing the constraints imposed by swamping. Additionally, Sakr et al. establish a connection between the decrease in accumulation precision and the training efficiency of DNNs by examining how the exacerbation of swamping phenomena, due to reducing accumulation precision, affects the variance of matrix multiplication outcomes (Sakr et al., 2019). Based on this analysis, they select an appropriate accumulation bit-width.

## 2.2 BFP FORMAT, BFP QUANTIZATION AND BFP MAC

BFP format is a numerical representation method wherein a group of data shares one exponent. Quantization methods that adhere to this data format can be classified as fixed-point uniform quantization (Jacob et al., 2018). Fixed-point uniform quantization can be categorized into multiple levels of methods based on the granularity of quantization. Quantization granularity varies, with per-tensor being the coarsest, using a single scaling factor for the entire matrix. Finer granularity is achieved through per-channel or per-token scaling. Block-wise quantization further refines this by dividing channels or tokens into blocks with a step size, yielding BFP quantization as a distinct variant with scaling factors as powers of two. Therefore, BFP quantization (Rouhani et al., 2023; Darvish Rouhani et al., 2023) can be expressed as:

$$\mathbf{X}_q = \lceil \frac{\mathbf{X}}{2^s} \rfloor, s = max(\lfloor \log_2^{|\mathbf{X}|} \rfloor) - N + 1 \tag{1}$$

where $\lceil \cdot \rfloor$ is the rounding function, $\mathbf{X}$ is the object to be quantized, $\mathbf{X}_q$ is the corresponding quantized result, $s$ is the scaling factor obtained through quantization, and $N$ is the number of bits used for the low-precision representation.

The BFP multiplier-accumulator architecture is bifurcated into two primary modules: the INT-MAC (Integer Multiply and Accumulate) and the FP-ACC (Floating Point Accumulate). The INT-MAC comprises a set of signed fixed-point multipliers, an addition tree, and an exponent summing adder, corresponding to the fixed-point multiplication and accumulation within the BFP inner product. This phase is termed **intra-block** computation. Conversely, the FP-ACC module includes normalization, an exponent alignment unit, an adder, and a fixed-to-floating-point conversion block, handling the floating-point accumulation of the BFP inner product. This stage is identified as **inter-block** computation. In Figure 1(d), we elucidate the implications of BFP format, intra-block and inter-block operations using a straightforward example. SE denotes the shared exponent, A and B represent the two matrices involved in the matrix multiplication computation, respectively, C denotes the resulting matrix, INT signifies the fixed-point result after intra-block fixed-point accumulation, and F indicates the number that has been normalized and is ready for floating-point accumulation.

## 3 MOTIVATION

### 3.1 HARDWARE BOTTLENECK ANALYSIS

The BFP MAC can be broadly categorized into fixed-point multiplication, fixed-point addition, and floating-point addition, corresponding to INT-MUL, INT-ACC, and FP-ACC as depicted in Figure

1(d). When weights and activations are quantized at a higher precision, INT-MUL constitutes the predominant area due to the inclusion of K (where K represents the block size) high-precision fixed-point multipliers. However, when weights and activations are quantized at an ultra-low precision, INT-MUL requires only ultra-low precision fixed-point multipliers, whereas the high-precision INT-ACC and FP-ACC become the primary area overhead. As illustrated in the Figure 1(b), in the BFP4 MAC with K=16, the area allocated to accumulation reaches 64.4%, indicating that reducing the precision of accumulation could yield significant hardware efficiency gains in this scenario.

## 3.2 MEAN, VARIANCE AND THE FROBENIUS NORM

Accumulation overflow is a critical issue to be addressed in the context of fixed-point quantization, which can have a significant impact on model performance. As shown in the Table 1, we observe that minor overflow rates cause slight performance decline, but increased rates lead to significant degradation in model performance. In the design of the MAC unit, it is common practice to calculate the theoretical maximum data range that the partial sums can reach based on the input data format to prevent overflow. Equation 2 is a formula for calculating the maximum bit width required for the partial sums based on the input data format. Here, both A and W are signed numbers.

$$K(2^{min(A_{width}-1, W_{width}-1)} - 2^{A_{width}+W_{width}-2}) \leq Partial\,Sum \leq K2^{A_{width}+W_{width}-2} \quad (2)$$

In deep learning models, partial sums rarely reach the theoretical extreme values because it is nearly impossible for all input tensors to be quantized to the extreme values. Consequently, the range derived from Equation 2 typically exceeds the actual data distribution. By the $3\sigma$ principle, the vast majority of data falls within $(\mu - 3\sigma, \mu + 3\sigma)$. Thus, bounding the partial sums by their mean and variance can mitigate data range wastage.

In the inference phase of deep learning models, the FP32 precision matrix multiplication is regarded as the benchmark for state-of-the-art performance. The inference quality is inferred to be superior when the outcomes of matrix multiplications using alternative precisions are closer to the FP32 results. Consequently, the challenge of correlating data precision with model accuracy can be reframed as one of determining the proximity between the reduced-precision result matrix and the FP32 precision result matrix. For this purpose, we focus on numerical approximation and employ the Frobenius norm (Suh et al., 2022; Yuan et al., 2020) as the metric for comparison.

Table 1: Average overflow rate for BERTbase in different accumulation widths and corresponding EM and F1-score on the SQuAD-v1.1 question-answering task

| Bit(A/W) | Accumulation Width | Average Overflow Rate | EM | F1 |
|----------|--------------------|-----------------------|--------|--------|
| 6/6 | 10 | 6.710% | 2.4976 | 10.939 |
| 6/6 | 12 | 0.017% | 75.639 | 83.938 |
| 6/6 | 24 | 0 | 78.978 | 86.667 |
| 8/8 | 14 | 7.894% | 2.6584 | 11.302 |
| 8/8 | 16 | 0.025% | 78.912 | 86.653 |
| 8/8 | 24 | 0 | 78.836 | 86.664 |

## 4 ACCUMULATION PRECISION ANALYSIS

In BFP format inner product computations, the process is divided into intra-block and inter-block stages. We ensure ample allocation for both the intra-block shared exponent width and the inter-block floating-point exponent width(We chose to allocate 8 bits like Microscaling(Rouhani et al., 2023)). Our research focuses on estimating the mean and variance of block-wise partial sums to determine the bit width for fixed-point multiplication and accumulation, and on relating the Frobenius norm to the mantissa precision of inter-block accumulations.

### 4.1 INTRA-BLOCK PARTIAL SUM MEAN AND VARIANCE ANALYSIS

Intra-block multiplication and accumulation refers to the process of performing multiplication and accumulation operations on weight elements($W_e$) and input elements($I_e$) that have been quantized using the BFP format. We note (with observations detailed in Appendix E) that the weights and

inputs participating in matrix multiplication are approximately distributed according to a Laplace distribution. To facilitate analysis, we hypothesize that the inputs conform to a Laplace distribution with a location parameter of 0 and a scale parameter of 1(The $W$ and $I$ below represent the original weights and inputs, respectively). Hence, we have $\mathbb{E}[I] = 0$. Furthermore, since BFP quantization is an unbiased estimator, it follows that $\mathbb{E}[I_e] = \mathbb{E}[I] = 0$. Additionally, $W_e$ and $I_e$ are independent of each other, and thus $\mathbb{E}[I_e \cdot W_e] = \mathbb{E}[I_e] \cdot \mathbb{E}[W_e] = 0$. Consequently, we can estimate the mean of the partial sums within the block to be 0. The variance calculation formula for the dot product terms within the block is as follows:

$$\mathrm{Var}[I_e \cdot W_e] = \mathbb{E}[I_e^2] \cdot \mathbb{E}[W_e^2] - \mathbb{E}[I_e]^2 \cdot \mathbb{E}[W_e]^2 \tag{3}$$

From the aforementioned analysis, we know that $\mathbb{E}[I_e] = 0$, thus we can express the variance as

$$\mathrm{Var}[I_e \cdot W_e] = \mathrm{Var}[I_e] \cdot \mathbb{E}[W_e^2] \tag{4}$$

According to the assumptions made in the preceding text, we can determine $\mathrm{Var}[I]$, $\mathbb{E}[W^2]$ and the mean of the shared exponent(How to calculate $\mathbb{E}[exp]$ is provided in the appendix A).

$$\mathrm{Var}[I_e] = \frac{\mathrm{Var}[I]}{2^{2(\mathbb{E}[I_{exp}] - bit + 1)}}, \quad \mathbb{E}[W_e^2] = \frac{\mathbb{E}[W^2]}{2^{2(\mathbb{E}[W_{exp}] - bit + 1)}} \tag{5}$$

With the mean and variance of the partial sums within the block, according to the $3\sigma$ principle, we consider each inner product term obtained from the intra-block inner product to fall within the range of $(-3\sigma, 3\sigma)$. Consequently, the range of the partial sums is $(-3K\sigma, 3K\sigma)$, where K is the number of terms in the sum. At this point, we can estimate the bit width required for fixed-point multiplication and accumulation. We have visualized the estimated bit width in the Figure 2.

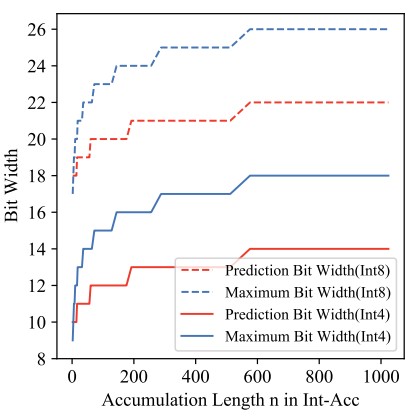

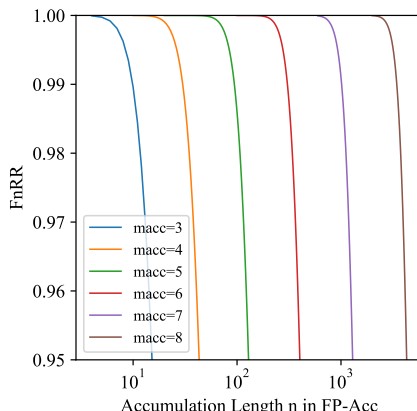

Figure 2: Intra-block Fixed-Point Accumulation Precisions for Llama2-7B

Figure 3: Intra-block Fixed-Point Accumulation Precisions for Llama2-7B

### 4.2 Inter-Block Accumulation Mantissa Precision Analysis

Let $p_i$ represent the i-th term for inter-block accumulation, $s_i$ denote the partial sum obtained from the i-th inter-block accumulation, $m_p$ and $m_{acc}$ correspond to the mantissa bit widths for $p_i$ and $s_i$, respectively, and $n$ denotes the length of the accumulation. Our key contribution lies in the proposal of a formula,

$$FnRR = \sqrt{\frac{\mathbb{E}[S_{n\,swamping}^2]}{\mathbb{E}[S_{n\,ideal}^2]}} \tag{6}$$

which correlates mantissa precision with model performance, where $FnRR$ is a function of $n$, $m_p$, $m_{acc}$, $\mathbb{E}[W]$, $\mathrm{Var}[W]$ and $K$, all precomputable parameters. In order to maintain performance under reduced precision, we aim for $FnRR \to 1$. As illustrated in the Figure 3, it can be observed intuitively that once $m_p, \mathbb{E}[W], \mathrm{Var}[W]$, and $K$ are determined, the $FnRR$ at a fixed mantissa precision is a waterfall-like curve with respect to the accumulation length $n$. The accumulation length for $FnRR$ is limited due to potential mantissa truncation caused by floating-point alignment during addition. This overflow leads to loss of significant digits, necessitating the introduction of

"swamping" to analyze its impact on $FnRR$ performance. As illustrated in the Figure 1(c), a single floating-point addition can be categorized into three scenarios: 1) "no swamping", which occurs when $|s_i| \leq 2^{m_{acc}-m_p}|p_{i+1}|$. 2) "full swamping," which occurs when $|s_i| > 2^{m_{acc}}|p_{i+1}|$. 3) "partial swamping," which occurs when $2^{m_{acc}-m_p}|p_{i+1}| < |s_i| \leq 2^{m_{acc}}|p_{i+1}|$. Subsequently, we will establish a connection between the Frobenius norm and the mantissa precision of inter-block summation from the perspective of swamping.

**Theorem 1.** The $FnRR$, Using $n$, $m_p$, and $m_{acc}$ to denote the accumulation length, the mantissa precision of accumulation terms, and the mantissa precision of the partial sum, respectively, $\sigma = \sqrt{K\text{Var}[I \cdot W]}$ where $K$ and $\text{Var}[W]$ are the block size for BFP quantization and the average variance of the weights selected from large models participating in quantization, is given as follows:

$$
FnRR = \sqrt{\frac{\sum_{i=1}^{n} P(A_i)\mathbb{E}[S^2_{i\,swamping}] + P(B)\mathbb{E}[S^2_{n\,swamping}]}{n\sigma^2}}
$$

$$
P(A_i) = \begin{cases} 2Q(\dfrac{2^{m_{acc}+1}}{\sqrt{2\pi}}), i = 1 \\ 2Q(\dfrac{2^{m_{acc}+1}}{\sqrt{2i\pi}})\prod_{j=1}^{i-1}(1 - 2Q(\dfrac{2^{m_{acc}+1}}{\sqrt{2j\pi}})), i = 2, 3, \ldots, n-1 \end{cases},
$$

$$
P(B) = \prod_{j=1}^{n}(1 - 2Q(\frac{2^{m_{acc}+1}}{\sqrt{2j\pi}})), \quad \mathbb{E}[S^2_{n\,swamping}] = n\sigma^2 - \sum_{i=1}^{n}\mathbb{E}[f_i^2],
$$

$$
\mathbb{E}[f_i^2] = \sum_{j=1}^{m_p} P(C_{ij})\mathbb{E}[f_{ij}^2], \quad P(C_{ij}) = 2(Q(\frac{2^{m_{acc}-j+m_p+1}}{\sqrt{2i\pi}}) - Q(\frac{2^{m_{acc}-j+m_p+2}}{\sqrt{2i\pi}})),
$$

$$
\mathbb{E}[f_{ij}^2] = \frac{2^{-2m_p-1}}{3}(2^j - 1)(2^{j+1} - 1)\mathbb{E}[2^{exp}]^2.
$$

(7)

The proof of this theorem is provided in the appendix B. Using Theorem 1, we endeavor to analyze the relationship between the precision of accumulation and the length of the cumulative process. When we set a very large $m_{acc}$, $P(A_i)$ will be close to 0, while $P(B)$ will be close to 1 and $\mathbb{E}[S^2_{n\,swamping}]$ will be close to $n\sigma$, which causes $FnRR \to 1$ as expected when the mantissa is maintained at high precision. When we set a very small $m_{acc}$, $P(B)$ will be close to 0, and $\mathbb{E}[S^2_{n\,swamping}]$ will be approximately equal to the sum of $P[A_i]\mathbb{E}[S^2_{i\,swamping}]$. When i is large, $P[A_i]$ will be close to 0. Consequently, in this case, $\mathbb{E}[S^2_{n\,swamping}]$ will be approximately equal to the sum of the first few terms of $P[A_i]\mathbb{E}[S^2_{i\,swamping}]$ when i is small. In other words, as $n$ increases, $\mathbb{E}[S^2_{n\,swamping}]$ will remain largely unchanged after an initial increase, leading $FnRR$ to rapidly approach 0 as $n$ increases. This indicates that with limited precision, there is little hope of maintaining computational accuracy when the length of accumulation is large. Similarly, because $FnRR$ exhibits a clear trend from 1 to 0 as $n$ increases at a fixed accumulation precision, $FnRR$ can provide a definitive decision boundary for the accuracy of accumulation.

### 4.3 PARAMETER SIGNIFICANCE ANALYSIS

Within Theorem 1, the computation of $FnRR$ is influenced by four parameters: $n, m_p, m_{acc}$, and $\sigma$, each exerting a distinct level of influence on the resulting calculation. Firstly, analyzing the parameter sigma reveals that $\mathbb{E}[2^{exp}]^2$ is approximately equal to $\sigma^2$, leading to $\mathbb{E}[S^2_{n\,swamping}] = f(n, m_p, m_{acc})\sigma^2$. Consequently, $\sigma$ has negligible impact on the computation of $FnRR$. Subsequently, we observe that the parameter $m_p$ is only employed in the calculation of $\mathbb{E}[f_i^2]$, and through scaling, we find that $\mathbb{E}[f_i^2] < \frac{\sigma^2}{6}$ (the proof of this conclusion is provided in the appendix C). Therefore, the parameter $m_p$ can, at most, decrease $\mathbb{E}[S^2_{n\,swamping}]$ to $\frac{5}{6}\mathbb{E}[S^2_{n\,ideal}]$, which in turn reduces $FnRR$ to around 0.913 at its lowest. The impact of $m_p$ on the computation of $FnRR$ is similarly insignificant. In summary, given a fixed mantissa precision, $m_{acc}$, $n$ is the predominant factor influencing the calculation of $FnRR$.

### 4.4 Mantissa Precision Analysis in Segmented Inter-Block Accumulation

As established in Section 4.3, the accumulation length $n$ is the most critical factor affecting the precision of inter-block accumulation. To achieve a lower inter-block accumulation precision while minimizing additional hardware overhead, a segmented approach to accumulation is adopted. Assuming $n = n1 \times n2$, the floating-point accumulation of length $n$ is segmented into $n2$ accumulations of length $n1$, which are then summed to yield the final computational result. Both segments of floating-point accumulation utilize the same mantissa precision to allow for the reuse of the floating-point addition unit. The proof of the formula is provided in the appendix D.

Theorem 2. Using a segmented accumulation method with $n = n1 \times n2$, where $n1$ is the segment length and $n2$ is the number of segments, the $FnRR$, with $m_p$ and $m_{acc}$ as the mantissa precision for the accumulation terms and partial sums, respectively, is provided in the subsequent sections:

$$FnRR_{segment} = FnRR(n1, m_p, m_{acc}, \sigma_{n_1}) \times FnRR(n2, m_{acc}, m_{acc}, \sigma_{n_2}) \qquad (8)$$

### 4.5 Usage of Theorem

We can ascertain the suitability of a certain inter-block accumulation precision by calculating its FnRR and evaluating its degree of convergence to 1, thereby predicting the most appropriate accumulation precision. The results indicate that when measured as a function of the accumulation length $n$ with a fixed precision, there exists a breakdown region for FnRR. This breakdown region is clearly observable when considering the normalized exponential loss:

$$f(n) = e^{n(1-FnRR)} \qquad (9)$$

In the Figure 4, we plot the $f(n)$ values at different inter-block accumulation precisions with ac-

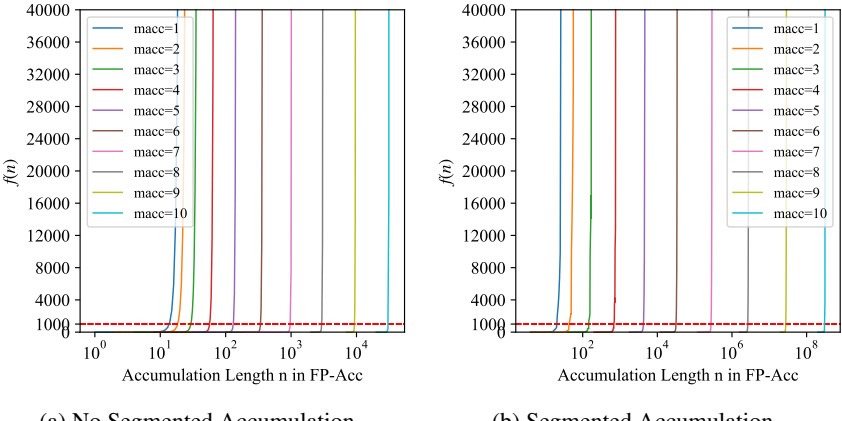

(a) No Segmented Accumulation  (b) Segmented Accumulation

Figure 4: (a) and (b) utilizes weight information from the Llama2-7B, with a block size of K equal to 32. The dashed line indicates the location of the breakdown point. It is readily apparent that below the dashed line, $f(n)$ rapidly approaches 1, whereas above it, $f(n)$ increases swiftly.

cumulation using segments of length 32 and no segmented accumulation. Here, we set $m_p$ to 9 (in practical applications, we can determine the corresponding $m_p$ value using the method described in section 4.1), and we use the weight data from Llama2-7B (Touvron et al., 2023) to calculate the FnRR. We can observe that $f(n)$ increases rapidly when it exceeds 1000, and it quickly approaches 1 when it is below 1000. Consequently, we select 1000 as the point of breakdown, such that accumulation precisions resulting in $f(n)$ values less than 1000 are considered suitable precisions.

## 5 Experiments

### 5.1 Experiment Setup

Through the aforementioned analysis, we predict the intra-block multiplication and accumulation bit widths, the inter-block accumulation mantissa bit widths, and the inter-block segmented accu-

mulation mantissa bit widths required for inference under different quantization configurations and segment lengths for the models (Llama2-7B, BERT-Large-Cased, BERT-Base-Cased, ResNet-50). We select these models due to their long accumulation lengths and because they belong to different applications, thereby enabling them to effectively validate our work. We aim to: 1) assess overflow occurrence in intra-block multiplication and accumulation at predicted precision, 2) evaluate and compare model performance with inter-block accumulation at predicted precision to FP32 baseline, and 3) evaluate and compare model performance with inter-block segmented accumulation at predicted precision to FP32 baseline. We employ MMLU (Hendrycks et al., 2020) testing to evaluate the performance of Llama2-7B, for BERT-Large-Cased and BERT-Base-Cased (Devlin et al., 2018), we use the SQuAD-v1.1 dataset (Rajpurkar et al., 2016) to finetuning and evaluate and for ResNet-50 (He et al., 2016), we use the CIFAR-10 (Krizhevsky et al., 2009) dataset to train and evaluate. Specifically, we utilize the Microsoft open-source MX Pytorch Emulation Library for quantization and choose 8-bit as the BFP quantization and accumulation exponent bit width.

To discuss the overflow situation of block-wise multiplication and accumulation and to implement the rounding of the partial sum during the inter-block accumulation process, we implement the BFP-format GEMM using PyTorch and CUDA, and we have inserted a rounding function at the location of partial sum accumulation to simulate the reduction in bit width.

## 5.2 OVERFLOW RATE IN INTRA-BLOCK OPERATIONS

We utilize the SQuAD-v1.1 to assess the model performance of BERT-Large and BERT-Base and the CIFAR-10 to assess the model performance of ResNet-50 following precision reduction. During inference, the matrix multiplication operations are then processed in BFP format, and the frequency of overflow events during computation is recorded to calculate the overflow rate. The results are presented in Table 2. BERT-Large and BERT-Base are evaluated using SQuAD-v1.1 across 48 topics, and the overflow rate is 0 in all cases. ResNet-50 is evaluated using CIFAR-10 and the overflow rate is also 0 in all cases. The experimental results confirm that no overflow occurs at the predicted fixed-point accumulation precision.

Table 2: The OR in this table represents the overflow rate. The data in the tuple is the result of BERT-Large and BERT-Base in SQuAD-v1.1 and ResNet-50 in CIFAR-10, respectively

| Precision | Block Size | Baseline Bit Width | Prediction Bit Width | Average OR |
|-----------|------------|--------------------|-----------------------|------------|
| BFP8 | 128 | 23 | 20 | (0,0,0) |
| | 64 | 22 | 20 | (0,0,0) |
| | 32 | 21 | 19 | (0,0,0) |
| BFP4 | 64 | 14 | 12 | (0,0,0) |
| | 32 | 13 | 11 | (0,0,0) |
| | 16 | 12 | 11 | (0,0,0) |

## 5.3 MODEL PERFORMANCE UNDER REDUCED INTER-BLOCK ACCUMULATION PRECISION

Table 3: The predicted inter-block accumulation bit width for our considered networks. Each table entry is an ordered tuple representing the bit widths for Llama2-7B, BERT-Large and ResNet-50, respectively. '-' signifies that we do not conduct tests on this quantitative configuration.

| Block Size | 8 | 16 | 32 | 64 | 128 |
|------------|-----|------|------|------|------|
| BFP4 | (-,7,-) | (7,6,6) | (7,5,6) | (6,5,5) | (-,-,-) |
| BFP4(Seg) | (-,4,-) | (5,4,4) | (4,3,3) | (4,3,3) | (-,-,-) |
| BFP8 | (-,-,-) | (-,-,-) | (7,5,6) | (6,5,5) | (5,4,4) |
| BFP8(Seg) | (-,-,-) | (-,-,-) | (4,3,3) | (4,3,3) | (3,2,2) |

The predicted bit width for each network and quantization precision are listed in Table 3 for the case of BFP and BFP segmented accumulation with the segment length calculated by $\sqrt{n}$. To elucidate that the inter-block accumulation precision identified by our method is precisely at the critical point, or as close as possible to the critical point while ensuring model performance (the critical point refers to the threshold at which a significant degradation in model performance is imminent), we evaluate the model performance under multiple sets of different accumulation precisions for each selected

model under various quantization configurations. Figure 7 reveals that as the accumulation precision decreases, there is a pronounced decline in model performance at the critical point. However, it is worth noting that when the accumulation precision is higher than the precision at the critical point, the change in model performance is not monotonic; it oscillates within a narrow range. This implies that there is no linear correlation between model performance and accumulation precision, as performance fluctuates around a certain level within a specific range of accumulation precision. When the accumulation precision is reduced below the critical point, there is a marked deterioration in model performance, which is consistent with the properties of FnRR.

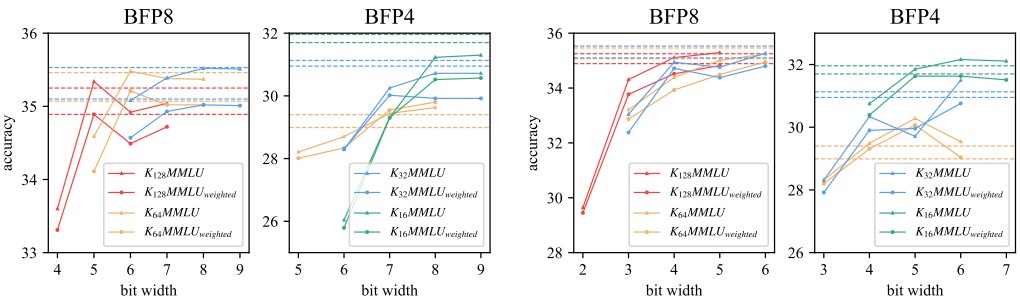

(a) No Segmented Accumulation results for Llama2-7B   (b) Segmented Accumulation results for Llama2-7B

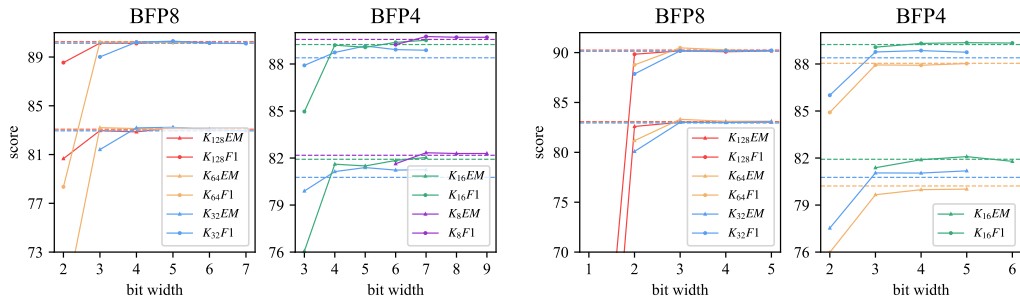

(c) No Segmented Accumulation results for BERTlarge   (d) Segmented Accumulation results for BERTLarge

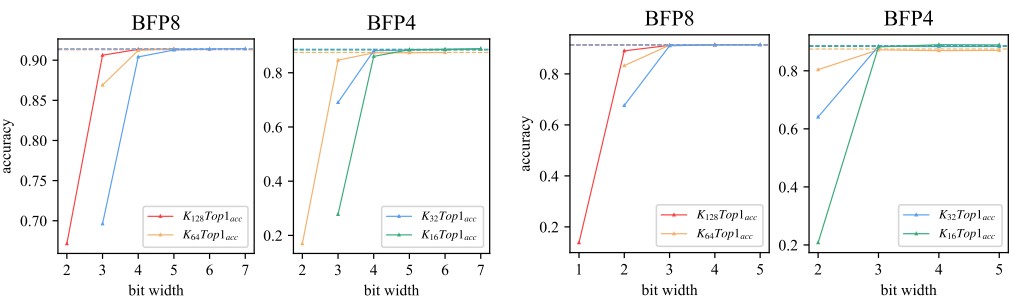

(e) No Segmented Accumulation results for ResNet-50   (f) Segmented Accumulation results for ResNet-50

Figure 5: The horizontal axis represents the inter-block accumulation precision, while the vertical axis indicates the score for the corresponding task. The dashed lines in the graphs denote the Baseline performance under the respective quantization configurations

### 5.4 MODEL PERFORMANCE UNDER REDUCED INTER-BLOCK SEGMENTED ACCUMULATION PRECISION

We select $\lfloor \sqrt{n} \rfloor$ as the segment length and evaluated the model performance under multiple sets of different accumulation precisions for each chosen model under every quantization configuration. Figure 7 demonstrates that as the accumulation precision decreases, there is a marked decline in model performance at the critical point. Furthermore, we can also find that employing segmented accumulation allows for at least a 1-bit reduction in precision while maintaining equivalent model performance compared to the no segmented accumulation method. In particular, the segmented

accumulation precision of 5 bits for BFP4 quantization of Llama2-7B with a block size of 16 outperforms the non-segmented method with a precision of 9 bits, achieving at least a 4-bit reduction. Both segmented and non-segmented methods at the predicted precision maintain performance close to the baseline, demonstrating the efficacy of our method in identifying minimal accumulation precision without substantial performance degradation.

## 5.5 HARDWARE IMPLEMENTATION

We utilize the formula derived in the preceding section to predict the accumulation precision for the Llama2-7B model with a block size of 16, for both BFP4 and BFP8 quantization precisions. The hardware design is completed based on the obtained accumulation precision, and we evaluate the area and power consumption using synthesis tools. As indicated in the evaluation, in terms of area, the BFP4 and BFP8 quantization precisions result in reductions of 28.7% and 13.8%, respectively. Notably, the reduction in area for the FP-ACC and Other components is significant. However, the area optimization for the INT-MAC is not pronounced due to the multitude of multiplier units, which do not decrease in area with the reduction in accumulation precision. Regarding power consumption, the BFP4 and BFP8 quantization precisions lead to decreases of 25.2% and 13.7%, respectively. Additionally, compared to the BFP16 Baseline, our optimized implementation of the BFP MAC at lower precision achieves significant improvements in area and power consumption, reaching up to $10.3\times$ and $11.0\times$ respectively.

Table 4: Analysis of area and power with varying quantization precisions, with the bolded segment reflecting area and power data derived from hardware design utilizing formula-predicted accumulation precision, contrasted with the non-bolded segment which is based on conventional accumulation precision for hardware design.

(a) Area Analysis

| Quantization Type | INT-MAC ($\mu m^2$) | FP-ACC ($\mu m^2$) | Other ($\mu m^2$) | Total ($\mu m^2$) |
|---|---|---|---|---|
| BFP4 | 142.54 | 133.34 | 38.90 | 314.78 |
| | **126.20 (↓11.5 %)** | **74.80 (↓44.0%)** | **23.56 (↓39.4%)** | **224.56 (↓28.7%)** |
| BFP8 | 619.07 | 147.84 | 49.09 | 816.00 |
| | **584.47 (↓11.4%)** | **90.11 (↓39.0%)** | **28.93 (↓41.1%)** | **703.51 (↓13.8%)** |

| (b) Power Analysis | | (c) Comparison with the BFP16 baseline | | |
|---|---|---|---|---|
| Quantization Type | Power (mW) | Quantization Type | Area ($\mu m^2$) | Power (mW) |
| BFP4 | 0.2208 | BFP4 | **224.56** | **0.1652** |
| | **0.1652 (↓25.2%)** | BFP16 | 2311.6 (10.3×) | 1.8204 (11.0×) |
| BFP8 | 0.5933 | BFP8 | **703.51** | **0.5122** |
| | **0.5122 (↓13.7%)** | BFP16 | 2311.6 (3.29×) | 1.8204 (3.55×) |

## 6 CONCLUSION

We present an analytical approach to predict the optimal accumulation precision for BFP GEMM operations in deep learning inference, balancing performance with precision. Our experiments confirm that this precision is near the limit while maintaining comparable performance to the baseline. Additionally, we demonstrate the effectiveness of segmented accumulation in further reducing floating-point precision. An interesting phenomenon is observed, where the decline in model performance with decreasing accumulation precision varies under different quantization configurations. Notably, highly quantized models exhibit a lower robustness and are more susceptible to reaching the precision boundary. Therefore, incorporating the impact of quantization on model robustness into our theoretical analysis could further improve our theoretical framework. We believe that our work provides theoretical support for the design of MAC units in deep learning inference.

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

## A   THE CALCULATION METHOD FOR $\mathbb{E}[exp]$

Let $K, \mu, \sigma$ denote the block size, quantization precision, mean, and variance, respectively, of the matrix selected for BFP quantization. In the main text, we assume that the means of the matrices participating in quantization follow a laplace distribution. The event $A_{ei}$ is defined as having i out of $K$ numbers within a block whose exponent is e, while the exponents of the remaining numbers are all less than e.

$$L(x, \mu, \gamma) = \begin{cases} 0.5e^{\frac{x-\mu}{\gamma}}, x < \mu \\ 1 - 0.5e^{-\frac{x-\mu}{\gamma}}, x \geq \mu \end{cases} \tag{10}$$

$$P(A_{ei}) = C_K^i [L(\frac{-2^{e-1} - \mu}{\sigma}) - L(\frac{2^{e-1} - \mu}{\sigma})]^{K-i}$$
$$\times [L(\frac{2^{e-1} - \mu}{\sigma}) - L(\frac{2^e - \mu}{\sigma}) + L(\frac{-2^e - \mu}{\sigma}) - L(\frac{-2^{e-1} - \mu}{\sigma})]^i \tag{11}$$

$$\mathbb{E}[exp] = \sum_{e=-\infty}^{+\infty} [2^e \sum_{i=1}^{K} P(A_{ei})] \tag{12}$$

From Equation 12, $E[exp]$ can be calculated. Our experiments have shown that when $e \in (-\infty, -50) \bigcup (50, +\infty)$, $P(A_{ei}) \to 0$. Therefore, Equation (9) can be simplified to

$$\mathbb{E}[exp] = \sum_{e=-50}^{50} [2^{e-bit+1} \sum_{i=1}^{K} P(A_{ei})] \tag{13}$$

## B   PROOF OF THEOREM 1

First, we present the assumptions that will be utilized in the subsequent derivations.

**Assumption 1:** BFP quantization does not alter the mean and variance of the matrix and the inner product terms obtained within the block are assumed to be independently and identically distributed.

This assumption is made for the convenience of determining the variance and mean of the floating-point numbers involved in the inter-block accumulation.

**Assumption 2:** We assume that the accumulation process stops when the first full swamping event occurs.

When full swamping occurs, the partial sum becomes sufficiently large relative to the accumulation terms. Although it is possible to recover from the full swamping event, the impact on the result is negligible.

**Assumption 3:** We consider that each bit of the mantissa of $p_i$ and $s_i$ is equally likely to be either 0 or 1.

This assumption is made for the convenience of determining the impact of discarding partial mantissa precision on Frobenius norm.

In order to calculate $FnRR$, we first need to compute the Frobenius norm when swamping occurs.To discuss the impact of swamping events on the Frobenius norm, we define the event $A_i$ as the first occurrence of full swamping during the accumulation process at the i-th accumulation. This definition also implies that full swamping do not happen in the accumulations for $i = 1, 2, \ldots, i-1$. The event $A_i$ happens if

$$|S_i| > 2^{m_{acc}} |p_{i+1}| \ \& \ |S_{i'}| \leq 2^{m_{acc}} |p_{i'+1}|, i' = 1, 2, \ldots, i-1 \tag{14}$$

To calculate the probability of event $A_i$ occurring, we first need to determine the distribution of $S_i$ and $p_i$. $p_i$ represents the i-th term in inter-block accumulation, which is essentially the result of a single block-wise multiplication and accumulation. According to Assumption 1, we calculate that $p_i \sim \mathcal{N}(0, K\text{Var}[I \cdot W])$ based on the central limit theorem. Similarly, $s_i$ is the sum of $p_i$, thus $s_i \sim \mathcal{N}(0, iK\text{Var}[I \cdot W])$. In the subsequent proof, we denote $K\text{Var}[I \cdot W]$ as $\sigma^2$.

Next, we aim to calculate the mean of $|p_i|$ to facilitate the computation of the probability of event $A_i$ occurring.

$$\mathbb{E}[|p_i|] = \int_{-\infty}^{+\infty} |x| \frac{1}{\sqrt{2\pi}\sigma} e^{-\frac{x^2}{2\sigma^2}} \, d\boldsymbol{x} \tag{15}$$

From Equation 15, we can compute that $\mathbb{E}[|p_i|] = \frac{2\sigma}{\sqrt{2\pi}}$. Therefore, we can derive the formula for calculating the probability of event $A_i$ occurring.

$$P(A_i) = P(|S_i| > 2^{m_{acc}}\mathbb{E}[|p_i|]) \cdot \prod_{j=1}^{i-1} P(|S_j| \le 2^{m_{acc}}\mathbb{E}[|p_j|]) \tag{16}$$

$$P(A_i) = \begin{cases} 2Q(\dfrac{2^{m_{acc}+1}}{\sqrt{2\pi}}), i = 1 \\[3mm] 2Q(\dfrac{2^{m_{acc}+1}}{\sqrt{2i\pi}}) \prod_{j=1}^{i-1} (1 - 2Q(\dfrac{2^{m_{acc}+1}}{\sqrt{2j\pi}})), i = 2, 3, \ldots, n-1 \end{cases} \tag{17}$$

Next, we calculate $\mathbb{E}[S^2_{n\,swamping}]$. First, we observe that partial swamping is possible in every accumulation, and we define the event $C_{ij}$ as the occurrence of stage j partial swamping during the i-th accumulation. Thus, event $C_{ij}$ happens if

$$2^{m_{acc}-j+m_p}|p_{i+1}| < |S_i| \le 2^{m_{acc}-j+m_p+1}|p_{i+1}| \tag{18}$$

Similar to the method for calculating the probability of event $A_i$ occurring, we derive the formula for calculating $P(C_{ij})$ as follows:

$$P(C_{ij}) = 2(Q(\frac{2^{m_{acc}-j+m_p+1}}{\sqrt{2i\pi}}) - Q(\frac{2^{m_{acc}-j+m_p+2}}{\sqrt{2i\pi}})) \tag{19}$$

Subsequently, we discuss the loss in Frobenius norm caused by stage j partial swamping. According to Assumption 3, the probability of a truncated bit being either 0 or 1 is equal. Consequently, we can calculate the truncation loss $\mathbb{E}[f^2_{ij}]$ occurring at the i-th accumulation.

$$\mathbb{E}[f^2_{ij}] = 2^{-2m_p+2\mathbb{E}[exp']} \sum_{r=1}^{2^j-1} \frac{r^2}{2^j} \tag{20}$$

Here, $\mathbb{E}[exp']$ represents the mean of the exponent of $pi$, and its calculation method is similar to that of $\mathbb{E}[exp']$ and will not be elaborated further. Equation 20 can be simplified to:

$$\mathbb{E}[f^2_{ij}] = 2^{-2m_p+2\mathbb{E}[exp']-1} \frac{(2^j-1)(2^{j+1}-1)}{3} \tag{21}$$

After the aforementioned analysis, we can compute the loss $E[f_i^2]$ in the Frobenius norm caused by partial swamping at the i-th iteration and $\mathbb{E}[S^2_{i\,swamping}]$.

$$\mathbb{E}[f_i^2] = \sum_{j=1}^{m_p} P(C_{ij})\mathbb{E}[f^2_{ij}] \tag{22}$$

$$\mathbb{E}[S^2_{i\,swamping}] = i\sigma^2 - \sum_{l=1}^{i} \mathbb{E}[f_l^2] \tag{23}$$

We proceed to discuss the impact of full swamping on the Frobenius norm. As per Assumption 2, when full swamping occurs, the accumulation process is halted. This implies that if full swamping occurs during the i-th accumulation, then $\mathbb{E}[S^2_{n\,swamping}|A_i] = \mathbb{E}[S^2_{i\,swamping}]$. Furthermore, we must also consider the scenario where full swamping does not occur throughout the entire accumulation process. The event B is defined as the absence of full swamping in $n$ accumulations. Event B happens if

$$|S_i| \le 2^{m_{acc}}|p_{i+1}|, i = 1, 2, \ldots, n \tag{24}$$

$$P(B) = \prod_{i=1}^{n} (1 - 2Q(\frac{2^{m_{acc}+1}}{\sqrt{2i\pi}})) \tag{25}$$

In summary,

$$\mathbb{E}[S^2_{n\,swamping}] = \sum_{i=1}^{n} P(A_i)\mathbb{E}[S^2_{i\,swamping}] + P(B)\mathbb{E}[S^2_{n\,swamping}] \tag{26}$$

## C  THE CALCULATION OF THE UPPER BOUND OF $\mathbb{E}[f_i^2]$

As indicated by Equation 22, $\mathbb{E}[f_i^2] = \sum_{j=1}^{m_p} P(C_{ij})\mathbb{E}[f_{ij}^2]$. Firstly, we analyze $\mathbb{E}[f_{ij}^2]$, where we observe that $2^{\mathbb{E}[exp']}$ and $\sigma^2$ are approximately equal, thus leading to the conclusion that $\mathbb{E}[f_{ij}^2]$ will reach its maximum value $\frac{1-2^{-m_p-1}-2^{-m_p}+2^{-2m_p-1}}{3}\sigma^2$ at $j = m_p$. Therefore, we can infer that $\mathbb{E}[f_i^2] < \frac{\sigma^2}{3}\sum_{j=1}^{m_p} P(C_{ij})$. Furthermore, from Equation 19, we can deduce that $\sum_{j=1}^{m_p} P(C_{ij}) = Q(\frac{2^{m_{acc}+1}}{\sqrt{2i\pi}}) - Q(\frac{2^{m_{acc}++m_p+1}}{\sqrt{2i\pi}})$. Due to $\frac{2^{m_{acc}+1}}{\sqrt{2i\pi}} > 0$, then $Q(\frac{2^{m_{acc}+1}}{\sqrt{2i\pi}}) < \frac{1}{2}$. Therefore,

$$\mathbb{E}[f_i^2] < \frac{\sigma^2}{3}\sum_{j=1}^{m_p} P(C_{ij}) < \frac{\sigma^2}{6} \tag{27}$$

## D  PROOF OF THEOREM 2

As readily apparent from Appendix B, the Frobenius norm for an accumulation segment of length $n_1$ is $\mathbb{E}[S_{n_1\,swamping}^2]$. Let the variance of the data for an accumulation of length $n_1$ be denoted as $\sigma_{n_1}$. Then, the variance $\sigma_{n_2}$ of the data participating in the accumulation of length $n_2$ is $n_1\sigma_{n_1}^2[FnRR(n_1, m_p, m_{acc}, \sigma_{n_1})]^2$. Furthermore, since $\mathbb{E}[S_{n_2\,swamping}^2]$ can be approximated as $f(n_2, m_p, m_{acc})\sigma_{n_2}^2$.

Therefore, when employing segmented processing, the calculated result FnRR is:

$$
\begin{aligned}
FnRR_{segment} &= \sqrt{\frac{\mathbb{E}[S_{n_2\,swamping}^2]}{n_1 n_2 \sigma_{n_1}^2}} \\
&= \sqrt{\frac{f(n_2, m_p, m_{acc})n_1\sigma_{n_1}^2[FnRR(n_1, m_p, m_{acc}, \sigma_{n_1})]^2}{n_1 n_2 \sigma_{n_1}^2}} \\
&= FnRR(n1, m_p, m_{acc}, \sigma_{n_1}) \times FnRR(n2, m_{acc}, m_{acc}, \sigma_{n_2})
\end{aligned}
\tag{28}
$$

## E  APPLYING THEOREM TO TRAINING TASKS

We endeavor to apply our theoretical framework to training tasks. As illustrated in the Figure 6, we trained ResNet-18 on the CIFAR-10 image classification task with a block size of 128 under BFP8 quantization configuration for 90 epochs with a learning rate of 0.1. Given that the maximum accumulation lengths for ResNet-18 in forward, backward, and gradient computation matrix multiplications are 4608, 4608, and 131072, respectively, our theoretical analysis (Theorem 1) deduces that the corresponding floating-point accumulation mantissa widths for these three types of matrix multiplications are 4, 4, and 8 bits. We used the training results with FP32 accumulation as a baseline and conducted ablation studies on the forward floating-point accumulation mantissa width, backward floating-point accumulation mantissa width, and gradient computation floating-point accumulation precision mantissa width by controlling variables. The experimental results are depicted in the figure. Based on these results, we observed that reducing accumulation precision within an appropriate range does not affect the convergence of model training. Specifically, the accumulation precision for backward and gradient computation has a minimal impact on model convergence, while the forward accumulation precision has a relatively greater influence. The forward results serve as the foundation for gradient computation and backward propagation, demanding higher precision. Therefore, when intolerable loss occurs due to an overly small accumulation bit width, the model struggles to converge to a satisfactory local optimum. In summary, our experiment reveals that the data precision requirement for the forward process is higher than that for backward and gradient computation, thus validating the applicability of our theory in selecting accumulation precision for training tasks.

## F  THE EXPERIMENTAL RESULTS USING STOCHASTIC ROUNDING

In the image classification task on CIFAR-10, ResNet-18 exhibits an identical maximum accumulation length to that of ResNet-50. Consequently, the bit-width of the accumulation tail number for

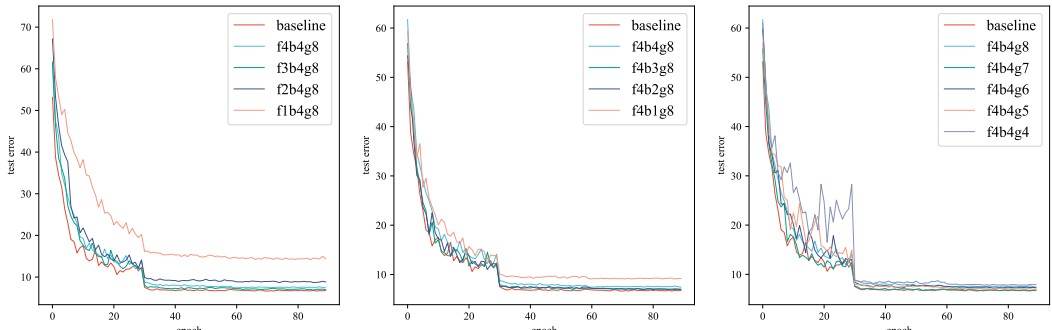

(a) The impact of forward bit width (b) The impact of backward bit width (c) The impact of gradient bit width

Figure 6: In the legend, fXbYgZ denotes the forward accumulation bit-width as X, backward as Y, and gradient as Z. For an instance, 'f4b4g8' signifies the training result curve obtained with a 4-bit forward accumulation bit-width, a 4-bit backward accumulation bit-width, and an 8-bit gradient computation accumulation bit-width.

ResNet-50, as presented in the Table 3, can be employed to deduce the corresponding accumulation precision for ResNet-18. The experimental outcomes are depicted in the Figure 7a, revealing a consistent trend between the quantization experiments utilizing stochastic rounding and those employing nearest rounding. Namely, as the accumulation precision diminishes, the model performance experiences a pronounced decline at a critical threshold.

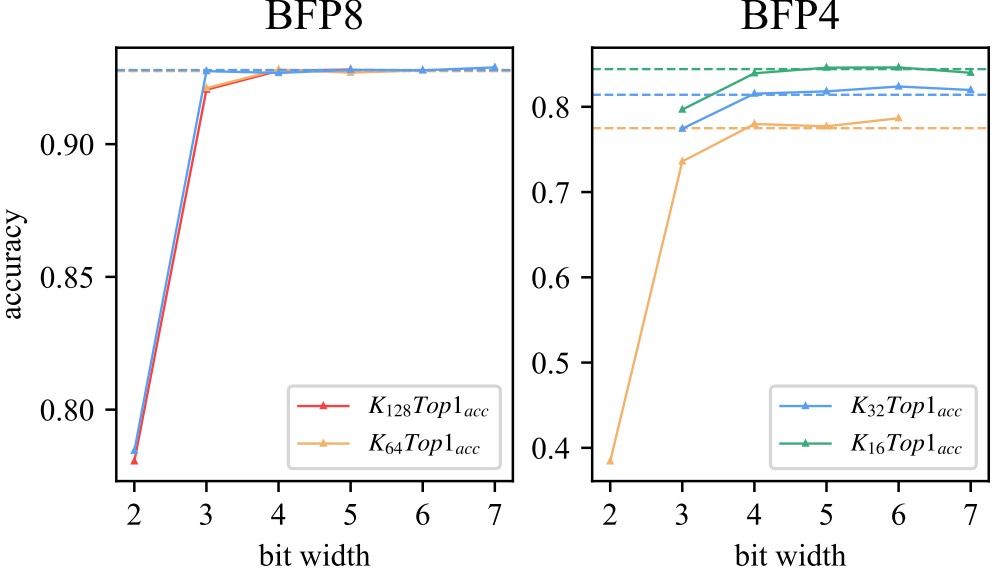

(a) No Segmented Accumulation results for ResNet-18 Using Stochastic Rounding

Figure 7: The horizontal axis represents the inter-block accumulation precision, while the vertical axis indicates the score for the corresponding task. The dashed lines in the graphs denote the Baseline performance under the respective quantization configurations

# G    DISTRIBUTION OF WEIGHTS AND INPUTS ENGAGED IN MATRIX MULTIPLICATION

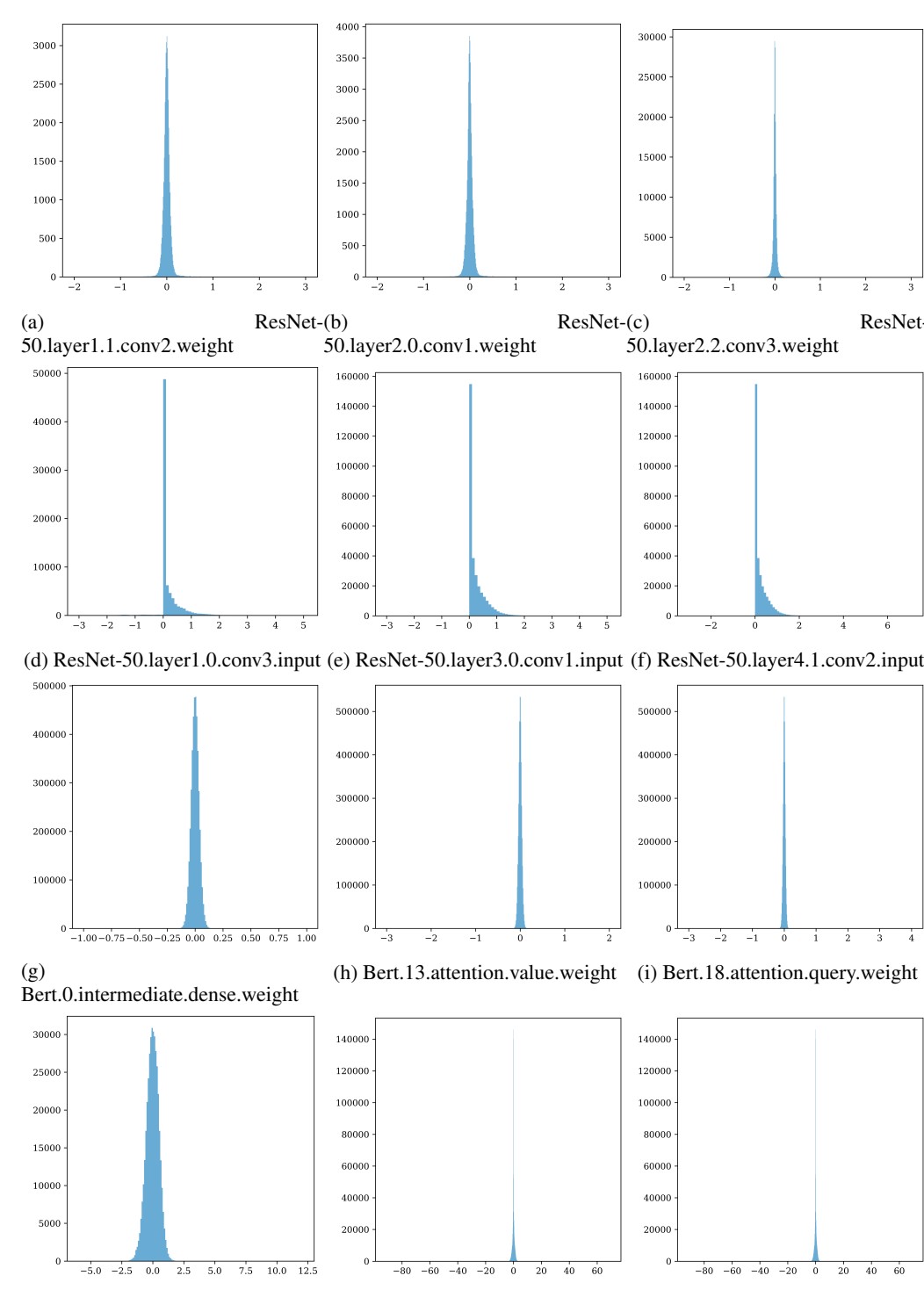

Figure 8: Each subplot visually represents the distribution of inputs and weights involved in matrix multiplication, randomly sampled from BERT-Large and ResNet-50, respectively.

# H EXPERIMENTAL DATA DETAILS

The following section provides detailed experimental results for the Llama2-7B model and the BERT-Large model.

Table 5: Experimental results of BERT-Large

| Accumulation Type | Quantization Precision | Block size | MACC | EM | F1 |
|---|---|---|---|---|---|
| Segmented | Int8 | 128 | 1 | 26.7833491 | 38.87238547 |
| | | | 2 | 82.59224219 | 89.83467333 |
| | | | 3 | 83.0179754 | 90.17436216 |
| | | | 4 | 82.99905393 | 90.0823776 |
| | | | 5 | 83.02743614 | 90.15831688 |
| | | | baseline | 83.07473983 | 90.24806709 |
| | | 64 | 2 | 81.1731315 | 88.76644966 |
| | | | 3 | 83.31125828 | 90.47422802 |
| | | | 4 | 83.11258278 | 90.26468859 |
| | | | 5 | 83.12204352 | 90.17531153 |
| | | | baseline | 83.00851466 | 90.14101486 |
| | | 32 | 2 | 80.08514664 | 87.85991477 |
| | | | 3 | 83.00851466 | 90.17600147 |
| | | | 4 | 82.98959319 | 90.17578653 |
| | | | 5 | 83.07473983 | 90.21952718 |
| | | | baseline | 82.9422895 | 90.13687096 |
| | Int4 | 64 | 2 | 75.97918638 | 84.91149044 |
| | | | 3 | 79.65941343 | 87.93871597 |
| | | | 4 | 79.98107852 | 87.91951769 |
| | | | 5 | 80.01892148 | 88.02584002 |
| | | | baseline | 80.21759697 | 88.04287399 |
| | | 32 | 2 | 77.5307474 | 86.00685635 |
| | | | 3 | 81.05014191 | 88.76773415 |
| | | | 4 | 81.04068117 | 88.85515301 |
| | | | 5 | 81.18259224 | 88.74837737 |
| | | | baseline | 80.76631977 | 88.3953108 |
| | | 16 | 3 | 81.38126774 | 89.07330911 |
| | | | 4 | 81.89214759 | 89.31319864 |
| | | | 5 | 82.09082308 | 89.35924713 |
| | | | 6 | 81.78807947 | 89.33690702 |
| | | | baseline | 81.9205298 | 89.24148335 |
| No Segmented | Int8 | 128 | 2 | 80.66225166 | 88.54197033 |
| | | | 3 | 82.96121097 | 90.13992802 |
| | | | 4 | 82.85714286 | 90.10981381 |
| | | | 5 | 83.19772942 | 90.27249646 |
| | | | 6 | 83.07473983 | 90.16693299 |
| | | | baseline | 83.07473983 | 90.24806709 |
| | | 64 | 2 | 68.17407758 | 78.34952856 |
| | | | 3 | 83.20719016 | 90.2162814 |
| | | | 4 | 83.13150426 | 90.22209627 |
| | | | 5 | 83.03689688 | 90.18009902 |
| | | | 6 | 83.05581835 | 90.15458593 |
| | | | baseline | 83.00851466 | 90.14101486 |
| | | 32 | 3 | 81.40964995 | 89.00883392 |
| | | | 4 | 83.18826868 | 90.22166107 |
| | | | 5 | 83.23557237 | 90.30179041 |
| | | | 6 | 83.0179754 | 90.15239831 |
| | | | 7 | 82.95175024 | 90.10275076 |
| | | | baseline | 82.9422895 | 90.13687096 |
| | Int4 | 32 | 3 | 79.89593188 | 87.91187527 |
| | | | 4 | 81.13528855 | 88.73624887 |

Table 5: Experimental results of BERT-Large

| Accumulation Type | Quantization Precision | Block size | MACC | EM | F1 |
|---|---|---|---|---|---|
| No Segmented | Int4 | 32 | 5 | 81.39072848 | 89.14670902 |
| | | | 6 | 81.22989593 | 88.91816976 |
| | | | 7 | 81.25827815 | 88.8771247 |
| | | | baseline | 80.76631977 | 88.3953108 |
| | | 16 | 3 | 76.06433302 | 84.96533694 |
| | | | 4 | 81.60832545 | 89.20004361 |
| | | | 5 | 81.49479659 | 89.06459835 |
| | | | 6 | 81.8448439 | 89.37870505 |
| | | | 7 | 82.03405866 | 89.51293669 |
| | | | baseline | 81.9205298 | 89.24148335 |
| | | 8 | 6 | 81.63670766 | 89.23406472 |
| | | | 7 | 82.33680227 | 89.75195921 |
| | | | 8 | 82.28949858 | 89.70555447 |
| | | | 9 | 82.28949858 | 89.70555447 |
| | | | baseline | 82.17596973 | 89.56697393 |

Table 6: Experimental results of Llama2-7B

| Accumulation Type | Quantization Precision | Block size | MACC | MMLU | MMLU-weighted |
|---|---|---|---|---|---|
| Segmented | Int8 | 128 | 2 | 29.64 | 29.45 |
| | | | 3 | 34.31 | 33.77 |
| | | | 4 | 35.11 | 34.52 |
| | | | 5 | 35.29 | 34.82 |
| | | | baseline | 35.25 | 34.89 |
| | | 64 | 3 | 33.21 | 32.87 |
| | | | 4 | 34.39 | 33.93 |
| | | | 5 | 35.01 | 34.49 |
| | | | 6 | 35.24 | 34.95 |
| | | | baseline | 35.46 | 35.07 |
| | | 32 | 3 | 33.04 | 32.38 |
| | | | 4 | 34.94 | 34.72 |
| | | | 5 | 34.76 | 34.38 |
| | | | 6 | 35.27 | 34.8 |
| | | | baseline | 35.53 | 35.1 |
| | Int4 | 64 | 3 | 28.31 | 28.2 |
| | | | 4 | 29.49 | 29.31 |
| | | | 5 | 30.28 | 30.07 |
| | | | 6 | 29.54 | 29.04 |
| | | | baseline | 29.4 | 28.99 |
| | | 32 | 3 | 28.31 | 27.92 |
| | | | 4 | 30.34 | 29.9 |
| | | | 5 | 29.71 | 29.96 |
| | | | 6 | 31.5 | 30.76 |
| | | | baseline | 31.13 | 30.95 |
| | | 16 | 4 | 30.75 | 30.39 |
| | | | 5 | 31.85 | 31.63 |
| | | | 6 | 32.16 | 31.63 |
| | | | 7 | 32.11 | 31.51 |
| | | | baseline | 31.96 | 31.7 |
| No Segmented | Int8 | 128 | 4 | 33.6 | 33.31 |
| | | | 5 | 35.34 | 34.89 |
| | | | 6 | 34.92 | 34.49 |
| | | | 7 | 35.04 | 34.72 |
| | | | baseline | 35.25 | 34.89 |
| | | 64 | 5 | 34.59 | 34.11 |

Table 6: Experimental results of Llama2-7B

| Accumulation Type | Quantization Precision | Block size | MACC | MMLU | MMLU-weighted |
|---|---|---|---|---|---|
| No Segmented | Int8 | 64 | 6 | 35.48 | 35.21 |
| | | | 7 | 35.38 | 35.02 |
| | | | 8 | 35.37 | 35.02 |
| | | | baseline | 35.46 | 35.07 |
| | | 32 | 6 | 35.08 | 34.57 |
| | | | 7 | 35.39 | 34.93 |
| | | | 8 | 35.52 | 35.02 |
| | | | 9 | 35.51 | 35.01 |
| | | | baseline | 35.53 | 35.1 |
| | Int4 | 64 | 5 | 28.21 | 28.01 |
| | | | 6 | 28.7 | 28.34 |
| | | | 7 | 29.44 | 29.55 |
| | | | 8 | 29.63 | 29.8 |
| | | | baseline | 29.4 | 28.99 |
| | | 32 | 6 | 28.29 | 28.3 |
| | | | 7 | 30.25 | 30.02 |
| | | | 8 | 30.72 | 29.92 |
| | | | 9 | 30.72 | 29.92 |
| | | | baseline | 31.13 | 30.95 |
| | | 16 | 6 | 26.04 | 25.79 |
| | | | 7 | 29.31 | 29.3 |
| | | | 8 | 31.23 | 30.52 |
| | | | 9 | 31.3 | 30.57 |
| | | | baseline | 31.96 | 31.7 |

Table 7: Experimental results of ResNet-50

| Accumulation Type | Quantization Precision | Block size | MACC | Top1 ACC |
|---|---|---|---|---|
| Segmented | Int8 | 128 | 1 | 0.138 |
| | | | 2 | 0.8905 |
| | | | 3 | 0.912 |
| | | | 4 | 0.9135 |
| | | | baseline | 0.9132 |
| | | 64 | 2 | 0.8324 |
| | | | 3 | 0.9127 |
| | | | 4 | 0.9126 |
| | | | 5 | 0.9141 |
| | | | baseline | 0.9135 |
| | | 32 | 2 | 0.6762 |
| | | | 3 | 0.9104 |
| | | | 4 | 0.9141 |
| | | | 5 | 0.9145 |
| | | | baseline | 0.9142 |
| | Int4 | 64 | 2 | 0.8039 |
| | | | 3 | 0.8722 |
| | | | 4 | 0.8693 |
| | | | 5 | 0.8705 |
| | | | baseline | 0.875 |
| | | 32 | 2 | 0.6407 |
| | | | 3 | 0.8847 |
| | | | 4 | 0.8832 |
| | | | 5 | 0.8837 |
| | | | baseline | 0.8868 |
| | | 16 | 2 | 0.2078 |
| | | | 3 | 0.8823 |

Table 7: Experimental results of ResNet-50

| Accumulation Type | Quantization Precision | Block size | MACC | Top1 ACC |
|---|---|---|---|---|
| Segmented | Int4 | 16 | 4 | 0.8898 |
| | | | 5 | 0.8891 |
| | | | baseline | 0.8841 |
| No Segmented | Int8 | 128 | 2 | 0.6714 |
| | | | 3 | 0.9062 |
| | | | 4 | 0.9135 |
| | | | 5 | 0.9135 |
| | | | baseline | 0.9132 |
| | | 64 | 3 | 0.869 |
| | | | 4 | 0.9122 |
| | | | 5 | 0.9142 |
| | | | 6 | 0.9134 |
| | | | baseline | 0.9135 |
| | | 32 | 3 | 0.6962 |
| | | | 4 | 0.9041 |
| | | | 5 | 0.9127 |
| | | | 6 | 0.9139 |
| | | | 7 | 0.9143 |
| | | | baseline | 0.9142 |
| | Int4 | 64 | 2 | 0.1694 |
| | | | 3 | 0.8465 |
| | | | 4 | 0.8736 |
| | | | 5 | 0.8738 |
| | | | 6 | 0.8748 |
| | | | baseline | 0.875 |
| | | 32 | 3 | 0.6905 |
| | | | 4 | 0.8813 |
| | | | 5 | 0.8845 |
| | | | 6 | 0.885 |
| | | | 7 | 0.8878 |
| | | | baseline | 0.8868 |
| | | 16 | 3 | 0.2776 |
| | | | 4 | 0.8604 |
| | | | 5 | 0.8851 |
| | | | 6 | 0.8868 |
| | | | 7 | 0.8887 |
| | | | baseline | 0.8841 |