# OpenReview forum: "Ultra-Low Accumulation Precision Inference with Block Floating Point Arithmetic"
_ICLR.cc/2025/Conference — Submitted to ICLR 2025_

### Official Review · Reviewer_wP2n · 2024-10-28

**Soundness:** 2
**Presentation:** 1
**Contribution:** 1
**Rating:** 3
**Confidence:** 2

**Summary:**

In the block floating point quantization, this paper proposes a statistical method to analyze the impact of reduced accumulation
precision on the inference of deep learning applications,
where formulates a set of equations to relate the data range of fixed-point
multiply-accumulate operations and the effects of floating-point swamping.
The experimental results show that area costs and hardware efficiency can be achieved,
demonstrating significant area reduction and power reduction.

**Strengths:**

This paper shows a contribution of arithmetic for low-cost inference.

**Weaknesses:**

Although this paper shows a contribution of arithmetic for low-cost inference,
I have several serious concerns as:

1. The main idea is not explicitly shown. I think that a metric FnRR denotes the ratio of floating point swamping.
The main idea should be focused on introduction and Figure 1, where the main idea or metric is not illustrated.
In the proof of Eq. (7), the normalized values based on assumption 1 in B. Proof of Theorem 1 are considered.
In the distributions of weights and activation, the assumption 1 is questionable. If there are any references for that, it could be better.

2. It is hard to understand this paper. Many italic and non-italic terms are mixed. For example, italic n and nonltalic n are used without discrimination.
Are sigma (line 316) and symbol sigma the same? Besides, too many other typos are shown.
In Figure 5, do terms in Y axis mean the accuracy on datasets? What is the meaning of scores?

3. The metric is only applied to inference. I think that this method can be evaluated on any training works.
I think that in the model for image classification, the proposed idea can be applicable to the model training. (ResNet 18 on CIFAR10 or ResNet50 on ImageNet-1K)
Besides, I think that resnet50 on CIFAR10 is not suitable for the job using floating-point format.

4. In hardware implementation, what is the environments for hardware synthesis?

In conclusion, in a point of arithmetic, the explanation should be more polished.
Besides, the effects of the proposed metric should be analyzed in other cases.

**Questions:**

Please, see the above weakness.

---

> ### Author Response · Authors · 2024-11-14
>
> Thank you for your careful review.
>
> For the first question, I present my main idea on lines 51 to 61. In short, there are both fixed-point and floating-point accumulations in BFP. Then I used variance and mean to estimate the data range of fixed-point accumulation based on the $3\sigma$ principle to determine the accuracy of fixed-point accumulation, and used FnRR to measure the degree of floating point swamping. Then based on the formula $f(n)$ in Section 4.5 and the threshold of 1000, we can determine the appropriate precision of floating-point accumulation. I also redrew Figure 1 to make its meaning clearer. Moreover, for assumption 1 in B, BFP quantization is a uniform linear quantization, and we complete the round by nearest rounding. It is worth noting that when the data distribution is close to symmetric, the nearest rounding is close to unbiased because it is symmetric in positive and negative errors. So we assume it's unbiased. In addition, according to our analysis of the importance of variables in Section 4.3, the variance of the data has little effect on the FnRR, so it is convenient for discussion that we make this assumption.
>
> As for the second question, we are very sorry for bringing you a bad reading experience. We have revised our paper and unified the symbols. The sigma(316 lines) and symbols $\sigma$ in the paper have the same meaning, which we have unified. Also, we redrew Figure 5, adding labels for the x and y axes. The scores in Figure 5 are representative of model performance.
>
> As for the third question, as you said, our method may be applied to training. We are doing experiments on training tasks and will submit the results as soon as we collect them.
>
> For the fourth problem, we use professional logic synthesis tools and open source 7-nm process Library (ASAP7) to synthesize.

---

> > ### Comment · Reviewer_wP2n · 2024-11-21
> > **Answer for the first rebuttal**
> >
> > Thanks for your answer.
> >
> > But there are still several concerns.
> >
> > Firstly, 3 $\sigma$ principle was assumed to determine the accuracy. Is it always valid in diverse models?
> >
> > Secondly, in your assumption, the unbiased data were assumed. Is this assumption also valid in any cases?
> >
> > Also, the revised manuscript has many typos. For example, 329, 333, 334 lines has non-italic terms: n1, n2, n3.
> > In Figure 4, what is the meaning "what is the meaning of f(n) rapidly approaches 1, whereas above it, f(n) increases swiftly"'
> > In line 276. I think that Theorem 1 is for FnRR in Eq. (7). Because this theorem only have equations, more explanation can be helpful.
> >
> > Considering the unsolved concerns, I will keep my rating at this time.

---

> > > ### Author Response · Authors · 2024-11-22
> > >
> > > Thank you for your reply.
> > >
> > > For the first question, we use the $3\sigma$ principle to judge the data range of fixed-point accumulation, according to which the precision of fixed-point accumulation is selected. When selecting the bit width of fixed-point accumulation, we select the data range upward (that is, to the wider bit width), so the actual range of supported data is greater than $3\sigma$. It is sufficient to satisfy the purpose of neither overflow nor waste of fixed point accumulation bit width. Our experiments also prove that the cumulative bit width of fixed points selected under various models will not overflow, that is, it will not cause model accuracy change.
> > >
> > > As for the second question, as we mentioned in the previous answer, the nearest rounding quantization method is unbiased for the mean value of data distributed with a symmetry of 0 values. When this distribution condition cannot be satisfied, we can also adopt the stochastic rounding method for quantization, and the unbias of random rounding has been proved in other work[1]. Whether the variance is unbiased or not does not affect the derivation of the formula (because if quantifying the difference brings a certain deviation $\Delta\sigma^2$, we can change the expression meaning of sigma in the formula from $KVar[I \cdot W]$ to $K(Var[I \cdot W]+\Delta\sigma^2)$), nor does it affect the final prediction result (for reasons analyzed in Section 4.3).
> > >
> > > As for the third question, we would like to thank you for pointing out our typos and we have corrected it. For Figure 4, this sentence means that the part below the dashed line, when $f(n)$ is less than 1000, its value approaches 1 rapidly as $n$ decreases, that is, $n(1-fnRR)$ approaches 0 rapidly, that is, $FnRR$ approaches 1 in this corresponding interval. The part above the dashed line, when $f(n)$ is greater than 1000, its value increases rapidly as n increases, that is, $FnRR$ moves away from 1 in this corresponding interval. According to our analysis in Section 4.2, we need to let FnRR approach 1 to maintain model accuracy, so we choose 1000 as the threshold for judgment. In addition, the key of theorem 1 lies in the calculation formula of $FnRR$, because the corresponding derivation process is too long to be written in the main text, so we have shown it in Appendix B.
> > >
> > > [1] Is Integer Arithmetic Enough for Deep Learning Training? NeurIPS, 2022

---

> > > ### Author Response · Authors · 2024-11-22
> > >
> > > We have included the experimental results regarding the reduction of accumulation precision following quantization via stochastic rounding in Appendix F for your reference. We look forward to your response.

---

> > > ### Author Response · Authors · 2024-11-27
> > >
> > > Have I solved your question? Looking forward to your reply.

---

> ### Author Response · Authors · 2024-11-15
>
> In the latest submitted version, I have modified some expressions in the introduction to better convey our main idea. I have mainly modified lines 60 to 80. Please kindly review and correct me.

---

> ### Author Response · Authors · 2024-11-17
>
> According to your suggestion, we applied our method to the image classification training task of ResNet18 on CIFAR-10. And we have added the results of the training task in the latest submitted paper. The newly added content is in Appendix E, please review it. According to our experimental results, in the training process, the data accuracy requirements of forward accumulation are the highest. In other words, according to our theory, the cumulative bit width selected for the forward process can also be applied to the reverse and gradient solutions.

---

> ### Author Response · Authors · 2024-11-21
>
> May I ask whether I have solved your question? Looking forward to your reply.

---

> ### Comment · Reviewer_wP2n · 2024-12-01
>
> Thank you for your responses to my concerns. As the discussion period is coming to an end, I have reviewed the revised PDF and your responses to other reviewers once again.
> Regardless of the contribution, the overall explanations in the paper make it very difficult for readers to follow. It is hard to enumerate all the issues. For example, the definition of a term in line 650 is incorrect. While the your response addressed this, Section 5.5 of the paper does not mention the process or tools related to power and area calculations. Besides, more explanations should exist for Figure 1.
>
> Thanks for your contribution again. However, considering the current state of the paper, I will reflect this in my rating.

---

> > ### Author Response · Authors · 2024-12-03
> >
> > Thank you for your review and valuable suggestions. Wishing you a joyful life.

---

### Official Review · Reviewer_nn8r · 2024-11-02

**Soundness:** 2
**Presentation:** 3
**Contribution:** 2
**Rating:** 5
**Confidence:** 4

**Summary:**

The paper aims to reduce the accumulator precision of the low-bit hardware matmul units where the accumulator becomes the hardware bottleneck. Based on the assumption that the inputs follow Laplace distribution, the paper analyzes the mean and variance for block floating-point format and proposes to use the mean with 3 standard deviations as the approximation of the largest magnitude that the accumulator should support, thus trimming down the accumulator precision. Experiments on ResNet 50, Bert-large, and llama2-7b shows the precision prediction fits well with the actual mininumal bits needed.

**Strengths:**

The paper shows clearly how the target of improvement and the bottleneck of accumulator in hardware in Figure 1, although more detailed description of the setup and source of the numbers shown in Figure 1(b) will be appreciated. The paper also clearly described its strategy, which is using the three standard deviations to estimate the largest output magnitude of the accumulator.

**Weaknesses:**

The main concern is on the experiments. Line 385 to 388 indicates that the evaluation mixes inference-only evaluation and training. It is very likely that the accumulator precision required in those two cases are very different. The accumulator precision for training can potentially be lower than inference-only approximation because overflow can serve as clipping, and model can still recover some quality through training. These two setups need to be separated and ablated.

In addition, Figure 5 is important as it shows how close the theoretical prediction matches the lower bound of the bitwidth needed for the accumulator in practice. However it is unclear in Figure 5 what the floating-point baseline is (dashed lines?). It is also unclear what the block size is. These are critical for assessing the experimental results.

**Questions:**

The questions are in the weakness section.

---

> ### Author Response · Authors · 2024-11-14
>
> In response to your main concern, in lines 385 to 388, I mentioned 3 groups of tasks, which are Llama2-7B for MMLU benchmark test, Bert-Large and Bert-Based SQuAD-v1.1 question-answering task, and ResNet-50 for image classification based on CIFAR-10 data set.  Among them, Bert-Large and Bert-Based, we first used the SQuAD-v1.1 data set to fine-tune the open source pre-training model in bert's official github repository in full precision, and then introduced quantization and reduced accumulation accuracy based on the fine-tuned checkpoint for question-answering test. The question-answering task process is the inference process. The same is true for Resnet50's image classification task based on the CIFAR-10 dataset. In addition, we did not plan to discuss the task of training at the beginning, because as you said, some overflow can be truncated in training but some quality can be recovered in subsequent training, but this is not included in our theory, which is very interesting and may be one of the contents of future research. In addition, I am also conducting experiments on training tasks, and I will submit the results as soon as I collect them.
>
> For Figure 5, which I have revised in the newly submitted paper and which I mentioned in the original on lines 474 to 476, the dashed line in the figure represents the baseline. In addition, block size is also marked in the legend. For example, in the legend on the left of Figure 5, it can be seen that the red line indicates that K is 128, that is, under the quantization configuration of block size 128, corresponding to the average accuracy of the accumulation percision obtained in the MMLU test benchmark.

---

> ### Author Response · Authors · 2024-11-15
>
> In the latest submitted version, I have modified some expressions in the introduction to better convey our main idea. I have mainly modified lines 60 to 80. Please kindly review and correct me.

---

> ### Author Response · Authors · 2024-11-17
>
> We have added the results of the training task in the latest submitted paper. The newly added content is in Appendix E, please review it. According to our experimental results, in the training process, the data accuracy requirements of forward accumulation are the highest. As you said, the cumulative bit width in the training task could be lower, and our experimental results coincide with your insight. This is interesting and worth exploring further. In other words, according to our theory, the cumulative bit width selected for the forward process can also be applied to the reverse and gradient solutions.

---

> ### Author Response · Authors · 2024-11-21
>
> May I ask whether I have solved your question? Looking forward to your reply.

---

> ### Comment · Reviewer_nn8r · 2024-11-26
> **Thank you for the answers.**
>
> I have read the response from the authors and other reviewers. I would like to keep the score.

---

> > ### Author Response · Authors · 2024-11-27
> >
> > Thank you for your review and valuable suggestions. Wishing you a joyful life.

---

### Official Review · Reviewer_Z6LM · 2024-11-02

**Soundness:** 3
**Presentation:** 2
**Contribution:** 3
**Rating:** 6
**Confidence:** 2

**Summary:**

Block Floating Point (BFP) quantization is introduced to improve the hardware efficiency in deep learning, but its accumulation logic becomes the hardware bottleneck especially for low-bit BFP quantization. This work studies the effect of reduced accumulation precision in BFP quantization and proposes a statistical  method to determine the appropriate accumulation precision. Experiments on Llama2-7B, BERT and ResNet-50 show that proposed approach can save 13%-28% area and reduce 13%-25% power while maintaining the model performance close to FP32 baseline.

**Strengths:**

+ This work presents a theoretical framework for analyzing the effect of accumulation precision in quantized GEMM, especially taking both data statistics and floating-point swapping into consideration. This provides a solid foundation for further research on quantization and its hardware design.
+ This work validates the proposed approach across different models and demonstrates the actual hardware benefits including area and power savings with a complete synthesized design.

**Weaknesses:**

- The proposed method relates the accumulation precision with the data range of actual workload, and thus predicts different accumulation precisions for different models. However, in real world, it is more common to run different models on the same hardware and thus it seems there is no need to specialize accumulation precision settings in hardware. Furthermore, if the hardware will be used for model training, the accumulation should also be able to handle the data range of model training, which is much larger than the inference. Therefore, it is doubtful if the proposed method is practical in the real world hardware design scenario.

**Questions:**

My questions are listed in the weakness section.

---

> ### Author Response · Authors · 2024-11-14
>
> Thank you for your affirmation and careful review of my paper.
>
> First, as you mentioned, in the real world, we need to run multiple models on one hardware. But this is not inconsistent with reducing the cumulative bit width. As our experiments show, our method can well predict the minimum accumulation accuracy required in the case of quantitative configuration determination. From Section 4.3, we analyze that the accumulation length n is the decisive factor in selecting the accumulation mantissa precision. Therefore, using our formula from another perspective, that is, we can calculate the maximum supported accumulation length by fixing the quantization configuration and the accumulation mantissa precision.
>
> For example, we use our formula to calculate that in a BFP quantization configuration with a block size of 128, a floating-point sum of 56 length can be supported with a precision of 4, corresponding to a matrix K dimension of up to 7168 participating in matrix multiplication; When the precision of the sum is 5, it can support the floating point sum of the sum length of 130, corresponding to the matrix K dimension of the matrix multiplication is up to 16640; It can be supported when the summation precision is 6, and the floating point summation length is 130, corresponding to the matrix K dimension participating in matrix multiplication is up to 43520. That is to say, in the case of only considering the inference task, the cumulative mantissa precision of 6 can support most applications.
>
> Second, for the training scenario you mentioned, first of all, on some end side devices we usually only need to deploy the model for inference tasks. Therefore, there are scenarios where our method can be used.
>
> Third, in response to your mention that the data range may be wider during training, this is not contrary to reducing the accuracy of the summations. Because the floating-point shortening mantissa widths cut off the lower bits, the absolute size of the data represented is much lower than the higher bits represented. For example, if the floating-point mantissa $(1.11001111)_{2}$ is truncated by the lower 4 bits, the missing data is only $0.05859375$, which is about $3.23\%$ of the original data. Therefore, I think it is also possible to reduce the accumulation accuracy during training. At the same time, I am also doing training experiments. After collecting the experimental results, I will submit them as soon as possible.

---

> ### Author Response · Authors · 2024-11-15
>
> In the latest submitted version, I have modified some expressions in the introduction to better convey our main idea. I have mainly modified lines 60 to 80. Please kindly review and correct me.

---

> ### Author Response · Authors · 2024-11-17
>
> We have added the results of the training task in the latest submitted paper. The newly added content is in Appendix E, please review it. According to our experimental results, in the training process, the data accuracy requirements of forward accumulation are the highest. Therefore, the design in real hardware according to the bit width of forward accumulation determined by our theory is sufficient to meet the bit width requirements of reverse and gradient solution. Therefore, our theory is also suitable for real hardware design scenarios.

---

> ### Author Response · Authors · 2024-11-21
>
> May I ask whether I have solved your question? Looking forward to your reply.

---

### Official Review · Reviewer_cr9Y · 2024-11-03

**Soundness:** 1
**Presentation:** 1
**Contribution:** 1
**Rating:** 1
**Confidence:** 4

**Summary:**

This paper investigates the impact of accumulator precision in BFP (Block Floating Point) hardware on the accuracy of neural networks. It provides separate analyses for the two types of accumulators in BFP hardware: intra-block and inter-block accumulators. Based on this accuracy analysis, the authors reduced the precision of BFP hardware, resulting in improvements in area and power efficiency.

**Strengths:**

Although the motivation behind this work is interesting and valid, the paper lacks sufficient detail and evaluation, making it difficult to clearly identify its strengths at this stage.

**Weaknesses:**

1.	The paper does not specify the format used for low-precision floating-point numbers. While FP32 is a well-known format with 1-bit sign, 8-bit exponent, and 23-bit significand, there is no standard format for floating-point numbers with fewer than 16 bits. For instance, FP8 can have either a 4-bit exponent and 3-bit significand or a 5-bit exponent and 2-bit significand. This paper does not clarify the specific format used for low-precision FP numbers.
2.	There is a lack of detail on the hardware implementation. The paper does not describe the hardware architecture considered, nor does it specify the bitwidth of the accumulators in both the proposed approach and the baseline.
3.	Although the paper provides an analytical approach to analyze the distribution of partial sums in Sections 4.1 and 4.2, there is no clear connection between this analysis and the optimization of accumulator bitwidth. Based on the results in Table 3 and Figure 5, and the fact that the impact of accumulator bitwidth varies across networks, the optimization of bitwidth appears to be empirical rather than directly derived from the analysis in Sections 4.1 and 4.2.

**Questions:**

1.	I suggest that the authors provide details on the low-precision floating-point formats used in this study.
2.	I recommend that the authors include more detailed information about the hardware implementation. For example, please provide a block diagram of the hardware and specify the bitwidth used for each component.
3.	In Equation (2), you mention that the range of partial sums depends on $2^{A_{width}}$ and $2^{W_{width}}$. However, it’s unclear whether the bitwidth refers to the exponent or mantissa, and it doesn’t specify whether it pertains to inter-block or intra-block partial sums, or the final accumulation results of the layer. If it refers to intra-block partial sums (as BFP only handles integer terms within the block), I believe the maximum bitwidth of the partial sum should be $log(k) + A_{width} + W_{width}$. Please clarify how you derived the term in Equation (2).
4.	In Section 4.1, what is the difference between $I_e$/$W_e$ and I/W? These terms are not clearly defined, making it difficult to follow the equations in this section.
5.	Please clearly label the x and y axes in Figure 5 for better interpretation.

---

> ### Author Response · Authors · 2024-11-13
>
> For question 1, the BFP data format mentioned in the paper is a blocky floating-point data, which is different from conventional floating-point data in that all elements in the block share an index instead of each data having its corresponding index. Generally, the shared index of BFP is 8 bits, and the bit width of each element is determined by the specific quantization configuration. BFP4 refers to an element with a bit width of 4bits, and BFP8 refers to an element with a bit width of 8bits. For example, a BFP8 data with a block size of 16 means that 16 data elements belonging to a block share an 8bit-sized index, and each of these 16 elements is stored in 8bits.
>
> For question 2, the hardware architecture of the BFP MAC is shown in Figure 1(a). According to your suggestion, I redrew Figure 1(a) and added the size of its usage bit width to each component. In baseline, we use FP32 as the floating-point cumulative bit width, and use the calculation formula you mentioned in question 3 to determine the fixed-point cumulative bit width, such as the fixed-point bit width we set $(\log_{2}^{16}+8+8)$ for a BFP8 MAC with block size 16.
>
> For question 3, the meaning of my formula and yours is the same, my formula determines the range of partial sum data representation, while your formula determines the bit width calculated according to the range, its representation meaning is the same. There are no exponents because we're talking about fixed-point numbers, and because we're talking about signed fixed-point numbers. Therefore, for a signed fixed-point number with a bit width of b, the representation range is $[-2^{b-1},2^{b-1}-1]$. In order to include all parts and data, bits are needed $\lceil log_{2}^{K}+log_{2}^{2^{A_{width}+W_{width}-2}+1}\rceil+1$, which is consistent with your formula.
>
> For question 4, I explained in line 214 of the paper that W and I refer to the elements corresponding to the weights and inputs quantized in BFP format, that is, the part of the data except the shared index. W and I know that the original weights and inputs are not quantized by BFP.
>
> For question 5, I have added the X-axis and Y-axis labels in a newly submitted PDF.
>
> For weakness 3, the bit widths in table3 are derived from the theories derived from Sections 4.1, 4.2 and 4.5. From 4.1, we can calculate the data distribution range through the $3\sigma$ principle, which is mentioned in line 232 to 236, to obtain the corresponding fixed-point cumulative bit widths. From 4.2 and 4.5, we can obtain the floating mantissa bit widths through the Equation 9 and $f(n)$ threshold 1000.

---

> ### Author Response · Authors · 2024-11-13
>
> Thank you for your recognition of my reply, and also thank you for your careful review and patient communication, which has provided me with great help to improve my work.
>
> As for the second point, I have modified the title in the original figure 5. Originally, I wanted to use INT4 and INT8 to represent the data types corresponding to BFP data elements, but I am very sorry for the confusion. So I changed the title correspondence to BFP8 and BFP4.
>
> As for the third point, Flexpoint and FAST as you mentioned did not use 8 bits as the shared exponential bit width, but in the latest related work such as Microscaling[1], 8 bits are generally used as the shared exponential bit width, and as I also mentioned in lines 207 to 208 of the paper, we allocated enough bit width to the shared exponential to simplify the discussion. As you suggested, I mentioned this point directly on lines 207 to 208. Referring to the bit width of FP32, we choose 8 bits as the bit width of the shared index after comprehensive consideration.
>
> As for the fourth point, thanks for your suggestion, I have revised it in line 218 in the newly submitted version.
>
> As for the fifth point, I realize that there is a clerical error, the correct formula should be 1, I have modified it in the newly submitted version, thank you very much for your reminding!
>
> As for the sixth point, I have described what a segment is in lines 327 to 332. As for how to select segments, our method does not restrict it, but we chose $\lfloor \sqrt{n}\rfloor$ as the segment length in line 481, that is, when we did the experiment. For an instance, if we have 100 numbers going into the floating point sum, then this is the sum length $n$ of 100, and with $\lfloor \sqrt{n}\rfloor$ as the segment length, we can get 10 floating point sum segments of length 10. Since both segmentation and unsegmentation are floating-point accumulators, this is a floating-point accumulator.
>
> As for the seventh point, I do not understand which picture you refer to in Figure 19(a). With respect to the inter-block accumulation accuracy, it refers to the accumulation accuracy of the FP ADDER in Figure 1(a).
>
> [1] With Shared Microexponents, A Little Shifting Goes a Long Way, ISCA, 2023.

---

> ### Author Response · Authors · 2024-11-15
>
> In the latest submitted version, I have modified some expressions in the introduction to better convey our main idea. I have mainly modified lines 60 to 80. Please kindly review and correct me.

---

> ### Author Response · Authors · 2024-11-17
>
> We have added the results of the training task in the latest submitted paper. The newly added content is in Appendix E, please review it.

---

> ### Author Response · Authors · 2024-11-21
>
> May I ask whether I have solved your question? Looking forward to your reply.

---

> ### Comment · Reviewer_cr9Y · 2024-11-25
>
> Although the authors' reply and the revisions to the paper have partially improved its quality, there are still too many unresolved issues to make the proposed solution compelling. Therefore, I will retain my original score.
>
> Key Issues:
>
> 1.	The explanations and definitions of several terms used in the paper remain unclear and insufficiently detailed. This lack of clarity makes it difficult to fully understand the methodology and its implications.
>
> 2.	I do not see the value or novelty of the proposed intra-block partial sum analysis and FnRR-based analysis. Both approaches utilize statistical properties of the layers, but the paper does not explain how these analyses offer any significant advantage over simpler and widely adopted methods such as min/max or 3-sigma-based truncation.
>
> 3.	The paper lacks a robust theoretical foundation to demonstrate how the proposed approach preserves accuracy. Despite this, it claims that the inter-block accumulation precision can be reduced to a bitwidth of 2–3 for BFP8 Seg (Table 3). This reduction seems overly aggressive and raises concerns about whether the baseline FP32 precision used in the comparisons is unnecessarily high, potentially skewing the evaluation.
>
> 4.	Figure 5 still lacks proper line descriptions, making it difficult to interpret the data presented.

---

> > ### Author Response · Authors · 2024-11-25
> >
> > As for the first question, I don't know what definition is not clearly expressed. Could you please give an example to illustrate it, because I have supplemented the definitions you mentioned before in the comment and the paper.
> >
> > For the second question, the analysis of the partial sum in the block is to determine a more suitable BFP MAC fixed point accumulation bit width. The $3\sigma$ principle is the analysis method we adopt, which can find a more suitable bit width compared with the maximum and minimum value to reduce the redundancy of the data range and achieve the purpose of simplifying the hardware. The analysis based on FnRR is to measure the impact of reducing the precision of floating-point accumulation mantissa on the data accuracy, so as to find a suitable floating-point accumulation mantissa bit width, and also achieve the purpose of simplifying the hardware. I have detailed how to build this connection in lines 60-80 of the paper and Section 4.2 and Appendix B. Using FnRR-based analysis, we can predict the appropriate floating point mantissa bit width in advance, and the hardware design based on this prediction can make the hardware area and power consumption smaller, which is also reflected in our experiment
> >
> > For the third question, the purpose of our method is to present a theorem to guide the selection of the appropriate accumulation bit width, because we find that there is a waste of high precision accumulation in the case of extremely low quantization accuracy. Through some statistical theorems and analysis, we derive a set of formulas for predicting the cumulative bit width. Our subsequent experiments also prove that the bit width predicted by our formula can maintain the model performance. We do not understand that the bit width precision reduction you mentioned is too aggressive. Therefore, both the theorem we deduced and the experimental results prove that in the BFP8(Seg) quantization configuration with a block size of 128, only 2 bits are indeed needed to maintain performance. In the conventional design, FP32 is used for accumulation, but we found that FP32 is not needed in fact, and the accumulation with low appropriate accuracy can also maintain the model performance, isn't it the significance of our work?
> >
> > For the fourth question, we analyzed Figure 5 in both Section 5.3 and Section 5.4. Through Figure 5 and Table 3, we can find that the accuracy of floating point summation mantissa predicted by us is close to the boundary of floating point summation mantissa accuracy that can keep the model performance from decreasing, thus proving the feasibility of our theory.

---

> > > ### Comment · Reviewer_cr9Y · 2024-11-25
> > >
> > > I do not intend to claim that the proposed method lacks merit, but the way the authors have presented the method is not sufficiently clear to convince me of its novelty and advantages.
> > >
> > > Key Concerns:
> > >
> > > 1.	Quality of Writing and Data Representation:
> > >
> > > The paper's explanations and data representations lack the clarity needed to effectively convey the novelty and contributions of the proposed method.
> > >
> > > 2.	Advancements Over Conventional Statistical Approaches:
> > >
> > > While the proposed analysis appears rigorous, it primarily relies on the statistics of inputs and weights, with no clear evidence demonstrating a direct correlation with model accuracy. Additionally, the method does not appear to be clearly distinct from partial sum precision lowering techniques based on naïve scanning or basic statistical measures such as standard deviation [1, 2]. Please note that partial sum lowering is a well-established concept in in-memory computing, where partial sum precision plays a critical role in efficiency. To establish the superiority of the proposed approach, the paper should explicitly demonstrate its advantages over naïve approaches. For instance, the paper could compare the correlation between the proposed statistical measures and model accuracy with those of conventional statistical measures, illustrating how the proposed analysis provides better guidance for precision reduction.
> > >
> > > 3.	Uncertainty About the Contributions to Bit Precision Reduction:
> > >
> > > While reducing bit precision as much as possible is important, it is unclear whether the extreme bit reductions achieved in this work are solely attributable to the proposed method. For example, the segmented approach appears to be a novel and meaningful contribution, as it improves precision reduction compared to the non-segmented approach. However, attributing the absolute reduction solely to the proposed method seems speculative without stronger evidence or analysis.
> > >
> > > 4.	Missing Clarifications on Proposed Methods:
> > >
> > > Certain aspects of the proposed methods lack sufficient detail. For example, the description of segmented inter-block accumulation is unclear. The paper states that inter-block accumulation involves summing FP-converted partial sums using an FP accumulator. However, what does "segmented inter-block accumulation" mean in this context? Does it imply that segments are accumulated using integers, or does it refer to an accumulation order where sums are computed within segments first and then aggregated across segments? Clarifying these details is essential for a complete understanding of the method.
> > >
> > > [1] Lee, Juhyoung, et al. "ECIM: exponent computing in memory for an energy-efficient heterogeneous floating-point DNN training processor." IEEE Micro 42.1 (2021): 99-107.
> > >
> > > [2] Sun, Hanbo, et al. "An energy-efficient quantized and regularized training framework for processing-in-memory accelerators." 2020 25th Asia and South Pacific Design Automation Conference (ASP-DAC). IEEE, 2020.

---

> ### Author Response · Authors · 2024-11-25
>
> First of all thank you for your detailed reply.
>
> As for the second point, because I am not familiar with the relevant research of in-memory computing, I did not compare it with it before. I read the two papers you pointed out. ECIM[1] experimentally found that when the selected cumulative bit width is 21, the result is measured under the $(1024\times1024)\times(1024\times1024)$ matrix multiplication initialized with the ResNet-18 training distribution. It is also mentioned that although increasing the length of the accumulation will make the method more profitable, a larger accumulation length will also introduce a larger calculation error, and it does not discuss the boundary of the accumulation length that the 21-bit width can support. Therefore, in real hardware design, if we need to bring each model into the data for calculation and then enumeration to choose the appropriate cumulative bit width, it will undoubtedly generate a huge amount of work. Our method uses statistics to predict the corresponding cumulative bit width in advance, which can save the above work. This is the point where our method is superior to ECIM, and the advantage of this prediction boundary is also explained in the abstract and Introduction, such as lines 15-19. In the second paper[2], the $3\sigma$ principle is also used to analyze the data range, but it analyzes the distribution range of quantified data, that is, it does not focus on the discussion of the accumulation part, which is inconsistent with our focus.
>
> For the third point, the partial sum analysis method and the floating-point accumulation precision analysis method based on FnRR proposed by me do not aim to further reduce the accumulation bit width, and we found that under extremely low quantization precision, the accumulation bit width can also be reduced, but how to determine the boundary of this reduction is still a challenge. That's why we propose these two methods. That is, to determine the boundaries that can be accumulated with reduced precision. The proposed  segmented approach is based on our analysis results in Section 4.3. In Section 4.3, we found that the length of accumulation is the decisive factor affecting the accuracy of accumulation. Therefore, in order to further reduce the accuracy of accumulation, we proposed the method of segmented accumulation, that is, changing the order of accumulation. As you said in point 4, accumulate in segments first, and then summarize across segments. We also extend our theory to the case of segmented accumulation in Section 4.4.
>
> For the fourth point, the segmented accumulation occurs in the floating-point accumulation, which is when the fixed-point part has been completed and converted to floating-point. And as I mentioned above, segmental accumulation actually changes the order of accumulation.
>
> [1] Lee, Juhyoung, et al. "ECIM: exponent computing in memory for an energy-efficient heterogeneous floating-point DNN training processor." IEEE Micro 42.1 (2021): 99-107.
>
> [2] Sun, Hanbo, et al. "An energy-efficient quantized and regularized training framework for processing-in-memory accelerators." 2020 25th Asia and South Pacific Design Automation Conference (ASP-DAC). IEEE, 2020.

---

### Meta-Review · Area_Chair_3F45 · 2024-12-19

**Metareview:**

This paper presents a statistical method to predict the boundaries of accumulation precision in deep learning inference using Block Floating-Point (BFP) arithmetic. The proposed approach aims to optimize hardware design by predicting the required accumulation precision, with a set of equations relating BFP quantization parameters to fixed-point multiply-accumulate operations and floating-point swamping effects. The method is validated on various models, demonstrating improvements in area and power efficiency while maintaining performance close to the FP32 baseline.

One of the primary concerns raised by the reviewers is the lack of clarity in the paper’s presentation. Another concern is the generalizability of the method, as some reviewers felt that it is tailored to specific models and may not scale well to other architectures or hardware configurations. Additionally, the contribution of the proposed method to bit precision reduction is unclear, and it is uncertain whether the extreme reductions in precision are primarily due to the new approach or if other factors, such as the segmented approach, play a significant role.

Reviewers had mixed opinions about the novelty and practicality of the proposed methods. This paper received an average score of 3.75, which is below the competitive threshold for this year’s submissions. Given the balance of strengths and weaknesses, the final recommendation is to reject this submission in its current form. The paper holds potential, but it would benefit from further revisions, including clearer explanations, a broader applicability study, and more detailed hardware implementation information.

**Additional Comments On Reviewer Discussion:**

During the rebuttal period, several key points were raised by the reviewers that provided further insight into the strengths and weaknesses of the paper.

Clarity of Methodology: One of the main concerns raised by Reviewer was the lack of clarity in terms and definitions, which impeded the understanding of the proposed method. The authors attempted to address this by providing additional explanations and clarifying certain terms. However, Reviewer remained skeptical, suggesting that the clarifications were still insufficient to fully resolve the ambiguity. Given that clarity is crucial for understanding the novelty and impact of the methodology, this concern weighed heavily in the final decision. Despite the authors’ efforts, the paper still lacked the necessary precision in presenting the core ideas.

Generalizability and Applicability: Reviewer raised concerns about the generalizability of the proposed method, especially its applicability beyond the specific models evaluated in the paper. The authors defended their approach, arguing that the methodology could be adapted to other models, though no additional experimental evidence was provided to substantiate this claim. While this point was partially addressed, the lack of broader validation meant that doubts about the method's scalability persisted. This concern was weighed as significant in the final recommendation, as the paper’s applicability to a wider range of models and hardware configurations remains uncertain.

Precision Reduction and Contribution: Reviewer questioned the contribution of the proposed method to bit precision reduction, suggesting that factors beyond the new approach might be influencing the results. The authors did not provide a clear explanation of whether the precision reductions were primarily due to their method or other aspects of the approach, leaving this question unresolved. This lack of clarity, along with the inability to distinguish the novel contributions, was another major point of concern. Given that this aspect was not sufficiently addressed in the rebuttal, it was weighed heavily in the final decision.

Hardware Implementation and Real-World Applicability: Reviewer pointed out the absence of detailed hardware implementation information and questioned the real-world applicability of the proposed method. The authors offered some additional insights into their hardware setup, but they did not provide enough concrete details regarding the synthesis environment or practical constraints. This concern remained unresolved in the rebuttal and was critical in the final assessment, as real-world validation is necessary to support the theoretical claims made in the paper.

In summary, while the authors made an effort to clarify certain aspects of their methodology and address reviewer concerns, several key issues remained unresolved. The paper still suffers from significant clarity issues, a lack of generalizability, and insufficient hardware implementation details. As a result, despite the authors' responses, the points raised by the reviewers were not adequately addressed, leading to a final recommendation of rejection.

---

### Decision · Program_Chairs · 2025-01-22

Reject